# Direct spinning and densification method for high-performance carbon nanotube fibers

Jaegeun Lee [1,4], Dong-Myeong Lee[1,2,4], Yeonsu Jung[3,4], Junbeom Park [1], Hun Su Lee[1], Young-Kwan Kim[1], Chong Rae Park [3], Hyeon Su Jeong [1] & Seung Min Kim [1]

Developing methods to assemble nanomaterials into macroscopic scaffolds is of critical significance at the current stage of nanotechnology. However, the complications of the fabrication methods impede the widespread usages of newly developed materials even with the superior properties in many cases. Here, we demonstrate the feasibility of a highly-efficient and potentially-continuous fiber-spinning method to produce high-performance carbon nanotube (CNT) fiber (CNTF). The processing time is <1 min from synthesis of CNTs to fabrication of highly densified and aligned CNTFs. CNTFs that are fabricated by the developed spinning method are ultra-lightweight, strong (specific tensile strength = 4.08 ± 0.25 Ntex$^{-1}$), stiff (specific tensile modulus = 187.5 ± 7.4 Ntex$^{-1}$), electrically conductive (2,270 S m$^2$kg$^{-1}$), and highly flexible (knot efficiency = 48 ± 15%), so they are suitable for various high-value fabric-based applications.

[1] Institute of Advanced Composite Materials, Korea Institute of Science and Technology (KIST), 92 Chudong-Ro, Bongdong-Eup, Wanju-Gun, Jeonbuk 55324, South Korea. [2] Department of Chemical and Biomolecular Engineering,  Korea Advanced Institute of Science and Technology (KAIST), 291 Daehak-Ro, Yuseong-Gu, Daejeon 34141, South Korea. [3] Department of Materials Science and Engineering, Seoul National University, 1 Gwanak-Ro, Gwanak-Gu, Seoul 08826, South Korea. [4] These authors contributed equally: Jaegeun Lee, Dong-Myeong Lee, Yeonsu Jung. Correspondence and requests for materials should be addressed to C.R.P. (email: crpark@snu.ac.kr) or to H.S.J. (email: jeonghs98@kist.re.kr) or to S.M.K. (email: seungmin.kim@kist.re.kr)

A carbon nanotube fiber (CNTF) is a macroscopic one-dimensional assembly of CNTs. So far, CNTFs have strength that is just a few percent of that of individual CNTs, because the properties of CNTFs are mainly determined by the interactions between CNTs, not by the nature of $sp^2$ C–C bonding within CNTs[1]. Nevertheless, theoretical calculation has predicted that the tensile strength of CNTF scales with the aspect ratio of constituent CNTs, as long as the intertube friction dominantly determines the strength of CNTF, and that achievable strength could be comparable to that of CNT if the constituent CNTs are perfectly aligned and densified along the fiber axis and have aspect ratio >$10^5$[2]. Interestingly, methods to produce highly aligned and densified CNTFs by wet fiber-spinning[3], and to produce CNTs with aspect ratio >$10^5$ by chemical vapor deposition (CVD)[4] have been reported, but CNTFs with comparable properties to CNTs have not been synthesized. The wet fiber-spinning method requires low-defect CNTs to form a liquid crystalline (LC) phase[5], but the defect density of CNTs with high-aspect ratio by conventional CVD methods is usually too high to meet this requirement[4]. A special synthesis method can produce CNTs with low defect density and high aspect ratio, but the method may not sometimes synthesize enough quantity of CNTs for fiber spinning[6]. As different approaches to synthesize high-performance CNTFs, various post-treatments have been performed on CNTFs that had been synthesized by the direct spinning method in which CNTFs are directly drawn from CNT aerogels formed in the CVD reactor. These treatments include the induction of chemical molecular crosslinking between CNTs[7,8]; liquid infiltration and subsequent densification[9–12]; polymer infiltration and sometimes subsequent carbonization[13]; mechanical densification[14,15]; and vapor phase carbon infiltration[16]. Most of the techniques successfully increase the mechanical or electrical properties, or both, of as-synthesized CNTFs, but in many cases these post-treatments require significant processing time to achieve meaningful improvement of CNTF properties. Therefore, the time scales of the post-treatments do not match that of the direct spinning, so they cannot be attached to direct spinning as a continuous process. So far, many of developed techniques that must be performed consecutively to produce high-performance CNTFs usually have their own limitations and are, therefore, sometimes mutually incompatible; this may be a reason that the techniques have not made significant impact in real industry.

The wet spinning method to produce CNTFs involves a process of forming LC phase of CNT solution in a superacid and usually produces highly aligned and dense CNTFs, because the characteristics of LCs enable CNTs to be highly concentrated and easily aligned by a shear force induced during the spinning of fibers. However, formation of a CNT LC phase entails a time-consuming mechanical stirring step to completely disperse CNTs in superacids as well as a purification step to remove amorphous carbon and residual catalysts (Fig. 1a)[17]. In contrast, the direct spinning method is a unique one-step process, in which CNTFs are synthesized and spun simultaneously in tens of seconds (Fig. 1b)[18]. However, internal structures of as-spun CNTFs are usually very porous when compared to CNTFs fabricated by wet spinning.

In this work, we present an optimized spinning method that can produce highly aligned and densified CNTFs in rapid and potentially continuous manner by combining the advantages of the wet spinning and direct spinning methods (Fig. 1c). When a directly-spun CNTF is immersed in chlorosulfonic acid (CSA), which is known as a true thermodynamic solvent for CNT dissolution[5], the CNTF swells as CSA penetrates it and protonates the CNTs. Appropriate stretching of the CNTF at this state rearranges entangled CNTs to improve their alignment in the axial direction. Then as the CNTF is immersed into a coagulation bath, the CSA is

extruded from the well-aligned CNTF by phase separation driven by solubility difference, to leave a highly packed CNTF with well-aligned structure. By applying this 1-min densification process for as-spun CNTF with high-specific tensile strength (2.1 N tex$^{-1}$) and low-defect density ($I_G/I_D$ ≈17), which can be obtained by the optimization of the direct spinning conditions, we successfully fabricate highly aligned and densified CNTFs with the specific tensile strength reaching up to 4.44 N tex$^{-1}$, and electrical conductivity up to 2270 S m$^2$ kg$^{-1}$. This work clearly shows that the optimization of the densification process for the proper degree of protonation (DOP) as well as the direct spinning conditions producing high-strength and low-defect as-spun CNTFs is critical to achieving the highest densification efficiency and thus leading to highly improved properties.

## Results

**Synthesis of high-strength CNTFs.** To synthesize high-strength as-spun CNTFs, we performed floating catalyst CVD (FC-CVD) using a vertical alumina tube reactor that had inner diameter of 85 mm and length of 1800 mm. The reactor was sealed by a water bath. Synthesized hollow CNT socks transformed to condensed fibers when they were drawn through the water bath. We optimized the direct-spinning conditions by fixing the total H$_2$ flow rate at 1200 sccm. This approach is in contrast to previous methods[19,20] in which H$_2$ flow rate or relative overall flow rate was adjusted. In our approach, the total H$_2$ flow rate was fixed based on the speculation that a convection vortex may form, but that a change of total H$_2$ flow rate could affect the vortex and thus affect the catalyst-formation process, so that optimal injection ratio of catalyst and carbon precursors could change. A recent simulation study[21] revealed that a convection vortex is developed at each of the entrance and exit of a system (horizontal or vertical), and that this vortex significantly affects the formation of CNT socks during a FC-CVD process. We roughly estimated the ratio of buoyancy force to viscous force which determines whether the backflow occurs in our vertical direct spinning system[22] and confirmed that most of our direct spinning conditions were within the range in which the backflow occurs.

At total H$_2$ flow rate of 1200 sccm and synthesis temperature of 1200 °C, we started the optimization process by adjusting the flow rates of thiophene and CH$_4$ with respect to the ferrocene flow rate (0.18 mg min$^{-1}$). The temperatures were 65 °C in the ferrocene container and −20 °C in the thiophene bubbler, and both catalyst precursors were supplied into the system by flowing H$_2$. The total H$_2$ flow rate was maintained at 1200 sccm by adjusting the additional H$_2$ flow rate. Decreased flow rates of CH$_4$ and thiophene tended to result in the synthesis of CNTFs that consisted mostly of CNTs with decreased numbers of walls; this result concurs with earlier reports[19,23]. However, with the fixed ferrocene flow rate (0.18 mg min$^{-1}$), we could only spin CNTFs that consisted mostly of double-walled CNTs (DWCNTs), even though the thiophene and CH$_4$ flow rates were decreased; further reduction in the relative ratios of thiophene and CH$_4$ failed to yield the amount of CNTs that is required to form CNT socks. Therefore, to synthesize CNTFs that consisted mainly of single-walled CNTs (SWCNTs), we increased the ferrocene flow rate to 0.3 mg min$^{-1}$ when we reduced the relative ratios of thiophene and CH$_4$. The types of CNTs that composed CNTFs were identified using transmission electron microscopy (TEM) and Raman spectroscopy (Fig. 2a–c). Synthesis of CNTFs that were composed mostly of SWCNTs, DWCNTs, and multiwalled CNTs (MWCNTs) required different synthesis conditions, and the CNTFs had distinct mechanical properties and characterization results (Fig. 2d–f, Table 1). Within the parameter space that we explored in this study, the synthesis conditions that produced

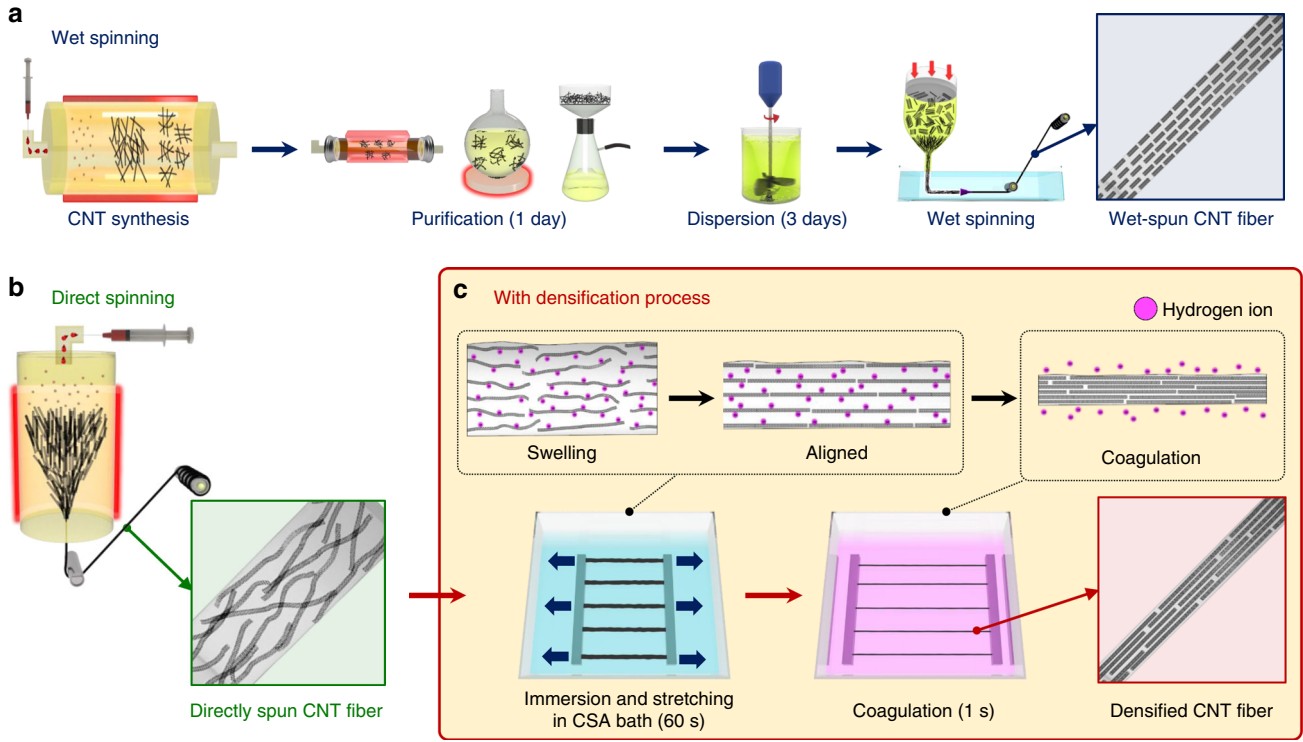

**Fig. 1** Schematic representation of spinning methods for CNTFs. **a** Wet spinning, **b** direct spinning, and **c** fast densification process based on the principles of the wet spinning method, which can be attached to the end of the direct spinning process

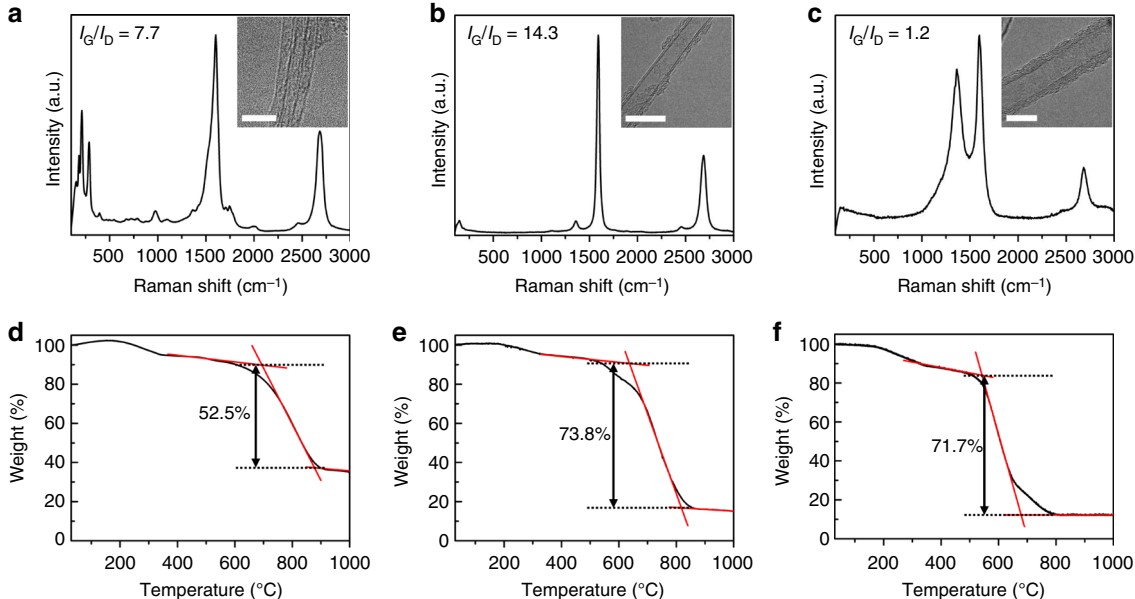

**Fig. 2** Characterization of as-spun CNTFs with different types of CNTs. Raman spectra of as-spun CNTFs dominantly consisting of **a** SWCNTs, **b** DWCNTs, and **c** MWCNTs with corresponding TEM images displayed as insets. TGA curves of as-spun CNTFs dominantly consisting of **d** SWCNTs, **e** DWCNTs, and **f** MWCNTs. The scale bars in the insets of **a**–**c** are 5, 10, and 10 nm, respectively

CNTFs that consisted primarily of DWCNTs corresponded to the conditions that produced the strongest CNTFs. It was reported that increase in the diameter of CNTs and reduction in the number of walls combine to increase contact, and thereby improve the mechanical properties of CNTFs[24]. Therefore, SWCNTs or DWCNTs are desirable constituents of CNTFs. However, as revealed by thermogravimetric analysis (TGA), in our synthesis conditions, the CNTFs that consisted primarily of SWCNTs had significantly larger amount of residual Fe than CNTFs that consisted primarily of DWCNTs (Fig. 2d, e, Table 1).

Finally, with a fixed ratio of carbon, sulfur, and iron that yields CNTF that consist primarily of DWCNTs, we slightly adjusted overall injection rates of $CH_4$, ferrocene, and thiophene with respect to the total $H_2$ flow rate to synthesize CNTFs that had the

**Table 1 Representative synthesis conditions and properties of various types of CNTFs**

|  |  | SWCNTF | DWCNTF | MWCNTF |
|---|---|---|---|---|
| Flow rates | $H_2$ (sccm) | 1200 | 1200 | 1200 |
|  | Methane (sccm) | 50 | 60 | 90 |
|  | Ferrocene (mg min$^{-1}$) | 0.3 | 0.18 | 0.18 |
|  | Thiophene (mg min$^{-1}$) | 0.17 | 0.58 | 0.83 |
| Atomic ratio | S: C | 0.00097 | 0.0028 | 0.0027 |
|  | S: Fe | 1.22 | 7.09 | 10.13 |
| Spinning rate (m min$^{-1}$) |  |  | 8–10 |  |
| Specific tensile strength (N tex$^{-1}$) |  | 1.15 ± 0.18 | 1.79 ± 0.22 | 0.79 ± 0.04 |
| TGA analysis | CNT (wt%) | 52.5 | 73.8 | 71.7 |
|  | Amorphous carbon (wt%) | 10.2 | 9.4 | 16.1 |
|  | Residual Fe (wt%) | 37.3 | 16.8 | 12.2 |

highest specific tensile strength. At each synthesis condition, we also optimized the spinning rate, which critically affected the final specific strength. Under the optimal synthesis condition, the alignment of as-spun CNTFs assessed by $I_{G\parallel}/I_{G\perp}$ (ratio of the intensity of G-peak ($I_G$) of Raman spectrum for the polarization parallel to the fiber axis to that for the polarization perpendicular to the fiber axis) increased with the spinning rate: from 24.6 ± 1.1 ($n = 5$) at the spinning rate of 5 m min$^{-1}$ to 35.1 ± 5.2 ($n = 5$) at the spinning rate of 9 m/min (Supplementary Fig. 1). The specific tensile strength increased from 1.14 ± 0.11 N tex$^{-1}$ ($n = 9$) with the linear density of 0.057 tex at the spinning rate of 5 m/min to 2.20 ± 0.14 N tex$^{-1}$ ($n = 10$) with the linear density of 0.044 tex at the spinning rate of 9 m/min (Supplementary Fig. 2). These results indicate that not only types of constituent CNTs, but internal structures of CNTFs are even more critical to the mechanical properties of CNTFs. The spinning rate higher than 9–10 m min$^{-1}$ did not further improve the mechanical properties of CNTFs in our case. We did not adjust the synthesis temperature and total $H_2$ flow rate, so the condition we found at the total $H_2$ flow rate of 1200 sccm and the synthesis temperature of 1200 °C might be a local optimum, not a global optimum; i.e., additional improvement may be possible.

**Synthesis of high-strength CNTFs with controlled $I_G/I_D$.** By principle, the efficiency of the CSA densification process should be highly dependent on the degree to which CSA solvated the CNTs in the CNTFs. CNTs are solvated by side-wall protonation by CSA[5,25,26] so the speed at which CSA penetrates the CNTFs can be strongly affected by the defect density of CNTs in them. To investigate how the defect density of CNTs in CNTF affects the densification process, we synthesized three CNTFs that had nearly identical tensile properties, but different defect density typically represented by $I_G/I_D$ from the Raman spectra of CNTFs. The intensity $I_D$ of the disorder peak in Raman spectra of CNTFs can be contributed by defects (e.g., kinks, Stone–Wales defects, and sp$^3$-hybridization) in CNTs that break the symmetry and perfection of sp$^2$-hybridized carbon network, and by carbonaceous impurities such as graphitic shells or particles in CNTF, which would not critically affect the solvation of CNTs in CNTF. However, the as-spun CNTFs synthesized in the optimized conditions consist mainly of CNTs (73.8 wt%) with a relatively small amount of carbonaceous impurities (9.4 wt%) (Fig. 2e, Table 1), so the correlation between the defect density ($I_G/I_D$ from the Raman spectra of CNTFs) and the solubility of CNTs composing CNTFs in CSA remains valid. $I_G/I_D$ of CNTF was controlled by slightly adjusting the ratio of CH$_4$ flow rate to the total $H_2$ flow rate. By adjusting the CH$_4$ concentration from 6 to 4 vol%, CNTFs having a low (5.23 ± 0.58, $n = 7$), medium (11.60 ± 1.89, $n = 7$), and a high (17.25 ± 3.99, $n = 7$) $I_G/I_D$ were synthesized. Regardless of $I_G/I_D$, the as-spun

CNTFs had high-specific tensile strength (>2 N tex$^{-1}$) and high-specific conductivity (>1700 S m$^2$ kg$^{-1}$, Supplementary Fig. 3); these are among the highest without any post-treatment process[20].

**Optimization of CSA densification process.** Starting with high-strength as-spun CNTFs that had medium $I_G/I_D$, we optimized the CSA densification process. Hereafter, CNTFs after the CSA densification process are referred to as "CSA-CNTFs". First, we examined the effect of the stretching ratio (the change in length divided by the original length and multiplied by 100) of the CNTFs on the specific strength while they were immersed in CSA for 1 min. As stretching ratio increased from 0 to 10%, the specific tensile strength steadily increased (Fig. 3a), but at stretching ratio higher than 10%, most of the CNTFs broke. Stretching ratio of 10% resulted in the highest improvement in specific tensile strength, without breaking the CNTFs. We also optimized the immersion time. The specific tensile strength of CNTFs increased as the immersion time increased from 0.5 to 1 min with the stretching ratio of 10%, but then decreased as the immersion time increased to 3 min (Fig. 3d).

The improvement in the specific strength of CNTFs depending on the stretching ratio of CNTFs during CSA densification process is correlated with internal structural changes of CNTFs: the alignment of CNTs along the fiber axis represented by $I_{G\parallel}/I_{G\perp}$ (Fig. 3b) and the volumetric density (Fig. 3c). For appropriate comparison, all real values (Supplementary Fig. 4) of specific strength, $I_{G\parallel}/I_{G\perp}$, and density were normalized to those of as-spun CNTF (Fig. 3). The method to measure the volumetric density is also described in the Methods section. At 0% stretching ratio, the specific strength and the density of CSA-CNTFs increased, but the alignment was slightly degraded; this result means that the specific strength is mainly increased by the increased density. As the stretching ratio increased, the specific strength and alignment increased monotonously, but the density did not change much; indicating that the specific strength was mainly affected by the improvement in the alignment (Fig. 3a–c). Therefore, the improvement in the specific strength of CSA-CNTFs can be explained by the interplay between the alignment and the density.

The existence of an optimal immersion time can also be explained by the change in alignment and density (Fig. 3d–f). Basically, a certain DOP is required for CSA to penetrate the CNTFs and to solvate CNTs by protonation. Alignment and density both increased as immersion time increased from 0.5 to 1 min; this result indicates that 0.5 min is too brief to cause sufficient structural rearrangement. However, excessive immersion time causes excessing DOP, so well-packed CNT bundles in the CNTF might disassemble. When the immersion time was further extended from 1 to 3 min, the alignment did not increase

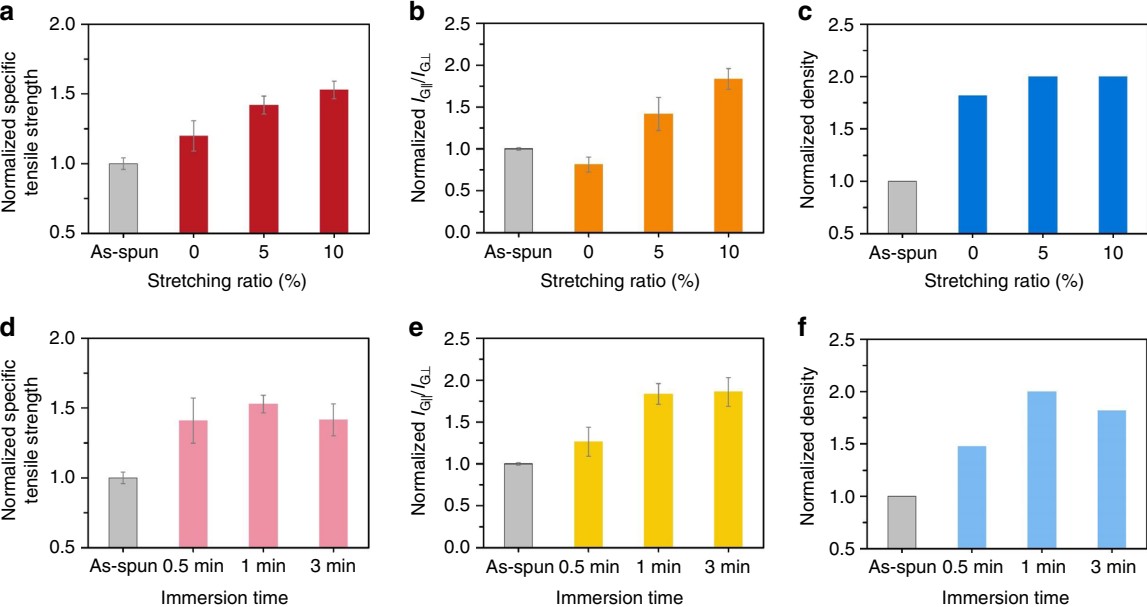

**Fig. 3** Optimization of CSA densification process. Evolutions in **a** normalized specific tensile strength, **b** normalized $I_{G\parallel}/I_{G\perp}$, and **c** normalized density of CNTFs having the medium $I_G/I_D$ during CSA densification process with various stretching ratios (immersion time, 1 min). Evolutions in **d** normalized specific tensile strength, **e** normalized alignment, and **f** normalized density of CNTFs with various immersion times (stretching ratio 10%). All values are normalized to those of as-spun CNTFs. Error bars represent the standard deviation. All real values are presented in Supplementary Fig. 4, including the numbers of replicates

and the density even decreased, causing overall decrease in the specific tensile strength of the CSA-CNTF. Thus, excessive immersion time is not helpful, and even might degrade the properties of the CSA-CNTFs.

**Improvement of the mechanical properties of CSA-CNTFs.** The optimized CSA densification process (immersion time: 1 min, the stretching ratio: 10%) was applied to as-spun CNTFs with different $I_G/I_D$ (Fig. 4a–c) and the mechanical properties of the CSA-CNTFs were significantly improved in all cases (Fig. 4d-i). Especially, the effect of CSA treatment was the most pronounced for CNTFs that had high $I_G/I_D$. In CNTFs that had low $I_G/I_D$, the specific tensile strength increased from $2.20 \pm 0.14$ N tex$^{-1}$ ($n = 10$) to $2.75 \pm 0.26$ N tex$^{-1}$ ($n = 9$) after CSA treatment. In CNTFs that had medium $I_G/I_D$, the specific tensile strength increased from $2.12 \pm 0.19$ N tex$^{-1}$ ($n = 12$) to $3.88 \pm 0.28$ N tex$^{-1}$ ($n = 15$), and in CNTFs that had high $I_G/I_D$, the specific tensile strength increased from $2.10 \pm 0.14$ N tex$^{-1}$ ($n = 12$) to $4.08 \pm 0.25$ N tex$^{-1}$ ($n = 10$). In all cases, the linear density changed negligibly, so the increase of the specific tensile strength was solely caused by the increase of the tensile load at failure. In general, an effect of a post-treatment is likely to be amplified when the properties of as-spun CNTFs are relatively low, because in these CNTFs, the room for the improvement is large. When the synthesis is well optimized, so the as-spun CNTFs are of high quality as in this work, efficiency of a treatment may not be so obvious. Nevertheless, the CSA treatment nearly doubled specific tensile strength and nearly quadrupled specific tensile modulus (from $48.3 \pm 7.4$ to $187.5 \pm 7.4$ N tex$^{-1}$) of CNTFs that had high $I_G/I_D$. The results indicate that the CSA densification can effectively maximize the mechanical properties of CNTFs as long as as-spun CNTF has a low defect density. The maximum specific tensile strength and modulus reached 4.44 and 195 N tex$^{-1}$, respectively. To the best of our knowledge, this is the world-best record among reported specific tensile strengths of CNTFs that have sufficiently long gauge length (2 cm); i.e., these CSA-CNTFs achieve both high strength and light

weight. In addition to the high-specific strength, the specific tensile modulus of CSA-CNTF is superior to those of most commercialized high-strength carbon fibers such as T1000G, which has specific tensile modulus = 163 N tex$^{-1}$ and the specific tensile strength = 3.54 N tex$^{-1}$[27].

The structural change of the CNTF after CSA treatment was analyzed using the CNTF with high $I_G/I_D$. The cross-sections of CNTFs before and after CSA treatment were observed using a scanning electron microscope (SEM) after cutting them using a focused ion beam (FIB) (Fig. 5a–d). After CSA treatment, the cross-sectional area was reduced (Fig. 5a, c) and the proportion of voids in the cross-section was significantly reduced (Fig. 5b, d). Accordingly, the density increased from 0.7 to 1.1 g cm$^{-3}$. To exclude the possibility that the densification can be partly induced by the acetone treatment, we also analyzed the cross-section and the density of CNTF before and after acetone treatment without the CSA stage; we observed no densification effect (Supplementary Fig. 5). Thus, the increase of density can be purely attributed to the CSA treatment process. The alignment assessed by $I_{G\parallel}/I_{G\perp}$ also increased after CSA treatment (Fig. 5e, f). These results prove that the CSA treatment effectively induces the rearrangement of CNTs as well as the densification. The structural changes of CNTFs with low and medium $I_G/I_D$s after CSA treatment were also analyzed (Supplementary Figs. 6 and 7). In these cases, the degree of densification and alignment after CSA treatment were also clearly increased. However, the final strengths of CSA-CNTFs can be affected by various factors such as the original internal structures and different interactions between CNT bundles as a result of different characteristics of the constituent CNTs. Therefore, these characterizations do not fully explain the different final strengths of CNTFs after CSA densification, which were synthesized in different conditions and thus had different defect densities, even though relative comparison among the samples after different treatments using the same original CNTF is quite useful as shown in Fig. 3.

The internal structural change of CNTF is also evidenced by the specific stress–strain curve; the curve itself does not clearly reflect

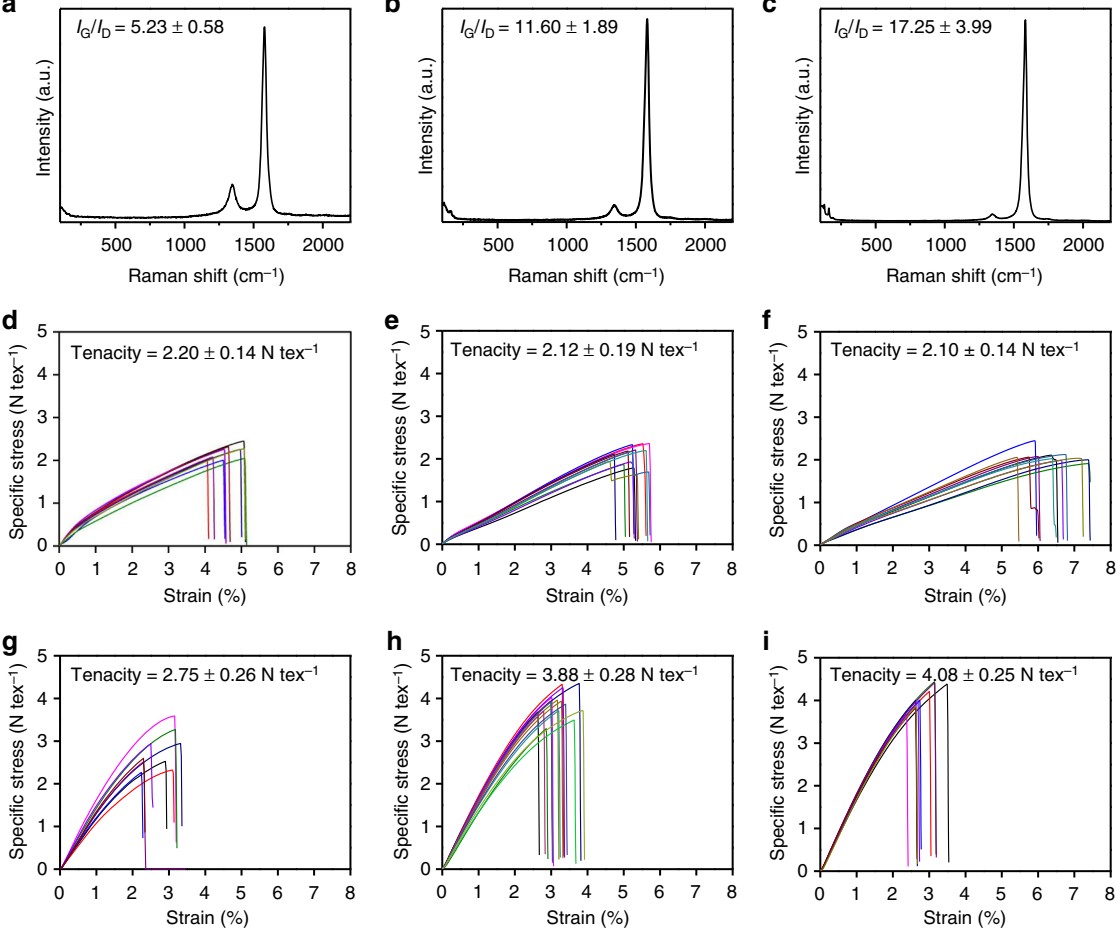

**Fig. 4** Effect $I_G/I_D$ of as-spun CNTFs on the CSA densification process. Raman spectra of as-spun CNTFs having a **a** low, **b** medium, and **c** high $I_G/I_D$. Specific stress–strain curves of as-spun CNTFs having a **d** low, **e** medium, and **f** high $I_G/I_D$. Specific stress–strain curves of CSA-CNTFs having a **g** low, **h** medium, and **i** high $I_G/I_D$

the differences in microstructures of CNTF between as-spun CNTF (Fig. 4f) and CSA-CNTF (Fig. 4i), but the specific modulus-strain curve, i.e., the first derivative of specific stress–strain curve with respect to strain, gives useful information on the microstructure of CNTF. The specific modulus-strain curves of as-spun CNTF (Fig. 5g) and CSA-CNTF (Fig. 5h) show a clear difference in their shapes. The specific modulus of as-spun CNTF initially decreases steeply, but at a certain point of strain shows a gradual decline, whereas that of CSA-CNTF decreases monotonically until it breaks. The two-mode decline of the modulus curve of as-spun CNTF implies a hierarchical structure of as-spun CNTF, which consists of numerous bundles and threads[28,29]. At low strain, the load is dominantly transferred to the thread-to-bundle structures. As the strain increases, the thread-bundle structures are sequentially broken, and the load is transferred to the remaining bundle-to-bundle structures. In contrast, the CSA-CNTF is packed tightly enough to reduce the degree of hierarchy in the structure of CNTF, so the failure occurs in more simplified way. We propose that this difference in modulus curves is clear evidence of the structural change after CSA treatment.

**Significance of DOP on improvement of the mechanical properties of CSA-CNTFs.** To increase understanding of the CSA densification process and the significance of DOP on the improvement of the mechanical properties of CSA-CNTFs, we also investigated the effect of oxidation on CSA treatment

efficiency, by using CNTFs with low $I_G/I_D$ ($\approx$5) and CNTFs with high $I_G/I_D$ ($\approx$17). Although defect density is the main factor determining the solubility of CNTs in CSA[5,25,26], it was recently revealed that the oxidation of CNTs dramatically increases their solubility in CSA, because the oxygen-containing groups can serve as centers for protonation[17]. Our hypothesis is that controlling the DOP is important to achieve highest efficiency of the CSA treatment; if we are correct, the effect of the oxidation on the efficiency of the CSA treatment would differ depending on the defect density in the CNTFs. Oxidation was conducted by heat treatment in air at 300 °C for 30 min. We first confirmed that the heat treatment in air did not significantly alter the defect density (Supplementary Fig. 8) and that the ratio of oxygen to carbon (O/C) after the heat treatment in air did not vary significantly among CNTFs with different $I_G/I_D$s (Supplementary Fig. 9).

When the $I_G/I_D$ was low, the CNTFs that had been heat-treated before CSA treatment (hereafter "heat-CSA-CNTFs") had higher specific tensile strength than CSA-CNTFs (Fig. 6a). In this case, the oxidation helped the CNTFs with low $I_G/I_D$ to have appropriate DOP, so the specific tensile strength of heat-CSA-CNTFs increased. As discussed in Fig. 3, the improvement of the mechanical properties after the CSA densification process can be explained by the interplay between the alignment and density improvement. When $I_G/I_D$ was low, the alignment of heat-CSA-CNTF was slightly lower than that of CSA-CNTF (Fig. 6b), but heat-CSA-CNTF had much higher density than CSA-CNTF (Fig. 6c); this difference seems to compensate for the slightly

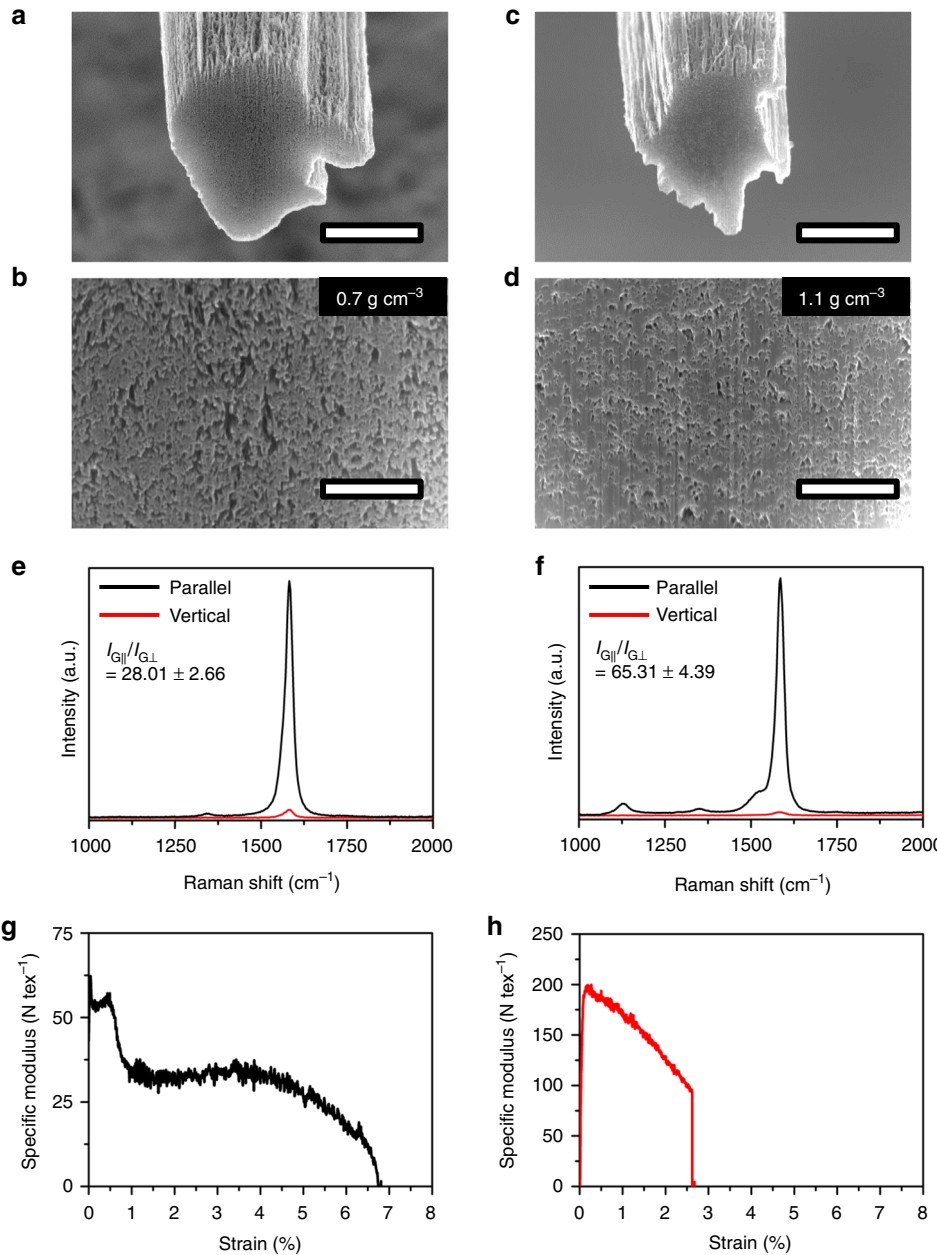

**Fig. 5** Structural change of CNTFs with high $I_G/I_D$ after CSA densification process. Cross-sectional SEM images of **a**, **b** as-spun CNTF and **c**, **d** CSA-CNTF. Polarized Raman spectra of **e** as-spun CNTF and **f** CSA-CNTF. Specific modulus-strain curves of **g** as-spun CNTF and **h** CSA-CNTF. The scale bars in **a** and **c** are 5 μm and those in **b** and **d** are 1 μm

lower alignment of heat-CSA-CNTF and results in overall improvement of specific tensile strength. On the contrary, when the $I_G/I_D$ was high, heat-CSA-CNTFs had lower specific tensile strength than CSA-CNTFs (Fig. 6a). In this case, the 1-min immersion in CSA would cause DOP to be too high; this condition is not desirable to achieve high specific tensile strength as discussed in Fig. 3d–f. The alignment of CSA-CNTF is much higher than that of heat-CSA-CNTF (Fig. 6b). Also, the density of CSA-CNTF is already much higher than that of as-spun CNTF, but heat-CSA-CNTF had a minor improvement in density compared to CSA-CNTF (Fig. 6c), so heat-CSA-CNTF had lower specific tensile strength than CSA-CNTF.

To confirm the hypothesis and compare the result with that in Fig. 3d–f, we performed CSA treatment with short immersion time (0.5 min) using heat-treated CNTF that had medium $I_G/I_D$ (Fig. 6d). The short immersion time yielded CNTF with increased

strength; this result indicates that the optimal immersion time for heat-treated medium-$I_G/I_D$ CNTF was shorter than 1 min, which was the optimal immersion time for medium-$I_G/I_D$ CNTF in Fig. 3d–f. This result, combined with the result in Fig. 3d–f, suggests that the optimal immersion time that yields optimal DOP is dependent on the properties of the CNTs that comprise the CNTF; when $I_G/I_D$ is relatively low, 1-min immersion is insufficient for the as-spun CNTF, but appropriate for the heat-treated CNTF, whereas when $I_G/I_D$ is relatively high, 1-min immersion is appropriate for the as-spun CNTF, but excessive for heat-treated CNTF.

**Discussion**

Our CSA-CNTFs have uniquely high flexibility, which is well represented by the high knot efficiency of $48 \pm 15\%$ ($n = 5$) (Fig. 7a). This value is lower than the knot efficiency of as-spun

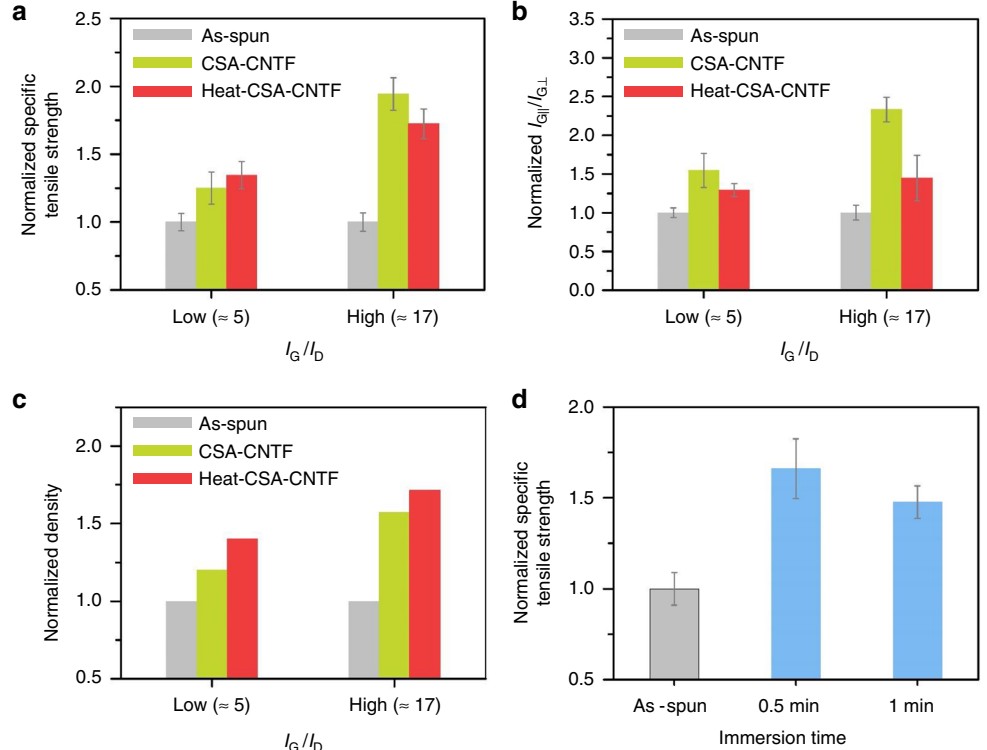

**Fig. 6** Combinational effect of $I_G/I_D$ and heat treatment. **a** Normalized specific tensile strength, **b** normalized $I_{G\parallel}/I_{G\perp}$, and **c** normalized density of as-spun CNTF, CSA-CNTF, and heat-CSA-CNTF with low and high $I_G/I_D$s. The immersion time and the stretching ratio in (**a**)–(**c**) were 1 min and 10%, respectively. **d** Normalized specific tensile strength of heat-CSA-CNTF that had medium $I_G/I_D$ as a function of immersion time. The stretching ratio in (**d**) was 10%. All real values of specific strength, $I_{G\parallel}/I_{G\perp}$, and density were normalized to those of as-spun CNTF. Error bars represent the standard deviation. All real values are presented in Supplementary Fig. 10, including the numbers of replicates

CNTF ($67 \pm 9\%$, $n = 7$), but much higher than that of high-strength carbon fibers[30]. Compared to the improvement of the specific tensile strength, the specific electrical conductivity showed a marginal improvement after CSA treatment (Supplementary Fig. 3), even though the specific electrical conductivity of our as-spun CNTFs is already comparable to that of metals (Fig. 7b). This lack of improvement implies that the specific electrical conductivity of CNTF cannot be improved above a certain point by only structural rearrangement of CNTFs, and may be restricted by the intrinsic properties of CNTs such as O/C or defect density ($I_G/I_D$). Therefore, the optimal internal structures as well as the characteristics of constituent CNTs for the best mechanical properties of CNTFs may be different from those for the best electrical properties of CNTFs. The specific tensile strength and specific electrical conductivity of the best CNTF sample were compared with those of various state-of-the-art high-performance fibers and metals from industry and research papers (Fig. 7b). Even the commercialized state-of-the-art materials do not possess both high specific tensile strength and specific electrical conductivity. However, our optimum CNTF has extremely high-specific tensile strength ($4.44 \text{ N tex}^{-1}$) as well as a sufficiently high-specific electrical conductivity ($2270 \text{ S m}^2 \text{ kg}^{-1}$), which makes our CNTF special.

The proposed method has the additional advantage of potential scalability. The time scale of each unit process is <1 min, so the CNTFs can be synthesized in a brief and potentially continuous process (Fig. 7c). The optimal immersion time is dependent on the property of as-synthesized CNTFs, so the length and design of the CSA bath should be determined based on the property of synthesized CNTFs as well as the optimized spinning rate of the scaled-up equipment. We have confirmed that the degree of

improvement largely depends on the defect density of CNTs; therefore, the key to the successful process lies in the synthesis of high-performance CNTFs that are composed of low-defect CNTs.

Development of CNTFs with comparable properties to individual CNTs is the ultimate goal of CNTF research. We proposed a highly efficient spinning method to fabricate CNTFs with comparable or even superior properties to those of commercialized state-of-the-art fibers and engineering metals, but our achievement still falls short of the ultimate goal. However, since our developed densification method allows CNTFs by the direct spinning to easily possess highly aligned and densified structures, the improvement of the direct spinning by the development of new catalyst precursors and thorough understanding of FC-CVD process for the synthesis of constituent CNTs with higher aspect ratio and low defect density can directly contribute to further improvement of CNTFs. This is because, without densified and aligned internal structures of CNTFs, improved properties of CNTs by the direct spinning method cannot be transferred to improved properties of CNTFs due to their hierarchical structures. Therefore, our study proposes an optimized direct spinning and densification method for high-strength CNTFs, and also provides a crucial research method for further improvement of CNTFs.

## Methods
**Synthesis of CNTF**. CNTF was continuously synthesized by a direct spinning method based on a floating catalyst CVD at 1200 °C. The reactor has an alumina tube whose inner diameter and length are 85 and 1800 mm, respectively. Ferrocene, thiophene, $CH_4$, and $H_2$ were used as a catalyst precursor, promotor, carbon source, and carrier gas, respectively. Ferrocene and thiophene were purchased from Sigma Aldrich (South Korea) and used as received without any purification. Each source was separately controlled and injected together into the tube at the top of

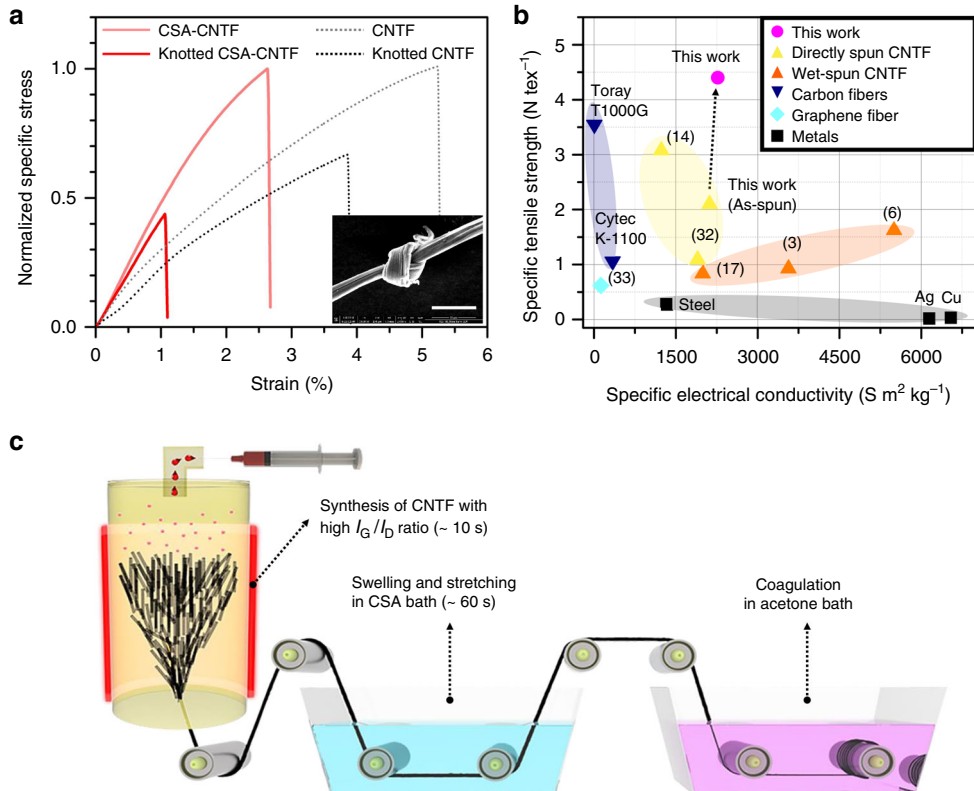

**Fig. 7** Excellent properties of densified CNTFs and proposed spinning method. **a** Normalized specific stress-strain curves of CNTF, knotted CNTF, CSA-CNTF, and knotted CSA-CNTF. **b** Ashby plot of specific tensile strength versus specific electrical conductivity of directly spun CNTFs (yellow triangles)[14,32], wet-spun CNTFs (orange triangles)[3,6,17], carbon fibers (navy inverted triangles)[27], graphene fiber (cyan diamond)[33], metals (black squares), and CNTFs from this work (magenta circle). **c** Proposed potentially continuous spinning method for highly densified and aligned CNTFs. The scale bar in (**a**) is 10 μm

the vertical furnace. Prior to the CNTF synthesis, the reactor was purged with high purity Ar (99.999%) for 5 min, and then the atmosphere was exchanged into $H_2$ (99.999%) immediately before the CNTF synthesis. Aerogel-like CNT assembly formed in vertical furnace was transformed into a fiber passing through a water bath. Synthesis conditions such as the ratio of the catalyst precursors (ferrocene and thiophene), $H_2$ flow rate, fiber spinning speed, etc. were finely adjusted to control the structure and morphology of CNTF. The ratio of the spinning rate to the feeding rate of catalyst precursors dominantly determined the alignment, packing density, and linear density of the CNTF. The CNTF was directly spun on a bobbin with the spinning rate from 5 to 10 m min$^{-1}$. To control $I_G/I_D$ of CNTF, the ratio of $CH_4$ flow rate to $H_2$ flow rate was finely adjusted. The $H_2$ flow rate of carrier gas (hydrogen) was kept constant at 1200 sccm and the concentration of $CH_4$ was slightly regulated within the range from 4 to 6 vol%.

**Densification of as-spun CNTF**. The CNTFs were cut into 15 cm long pieces and loaded on our custom-designed holder which is equipped with the screws to control the strain of the loaded CNTFs (Supplementary Fig. 11). The CNTFs were immersed in a CSA bath for 20 s and 10% strain was slowly applied in the following 40 s. Then, the CNFs were moved to an acetone coagulation bath. The CNTFs were further washed in water bath and acetone bath for two times and then dried in a vacuum oven. In some cases, the densified CNTFs were further dried in Ar atmosphere at 600 °C to completely remove any liquid trapped inside the CNTFs.

**Characterization of CNTFs**. The tensile properties and linear density of the CNTFs were measured using a FAVIMAT single-fiber tester (FAVIMAT-AIR-OBOT2, Textechno, Germany) using a 0–2 N load cell with a resolution of $10^{-6}$ N. The gauge length was 20 mm and the tensioning rate was 2 mm min$^{-1}$. Linear density was measured by the vibroscopic method with the pretension force of 0.07 cN following our proposed procedure for the accurate measurement[31].

The electrical conductivity was measured using a probe station (MST-4000A, MS Tech, Korea) by the four-point probe method with 20 mm distance between the electrodes.

The cross-sectional areas of CNTFs were obtained using ImageJ software from cross-sectional SEM images. For the observation, the CNTFs were cut using an FIB system (Helios, Thermofisher Scientific). Considering the tilt angle of the sample (52°), the cross-sectional area was calculated by dividing the area obtained from the

SEM image by cosine of 38°. Volumetric density of the CNTF was obtained by dividing the linear density by the cross-sectional area.

The structure of CNTs, the defect density of CNTFs and the contents of the residual catalyst particles and amorphous carbon in CNTFs were measured by using TEM (Tecnai G2-F20, Thermofisher Scientific), Raman (InVia, Renishaw), and TGA (Q50, TA Instruments) equipment, respectively.

## Data availability
The data that support the findings of this study are available from the corresponding author upon reasonable request.

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

## Acknowledgements
This study was supported by the grants from the Korea Institute of Science and Technology (KIST) Open Research Program. This work was also partly supported by the Industrial Fundamental Technology Development Program (10052838, development of the direct spinning process for continuous carbon nanotube fiber) funded by the Ministry of Trade, Industry and Energy (MOTIE), the Nano-Material Technology Development Program through the National Research Foundation of Korea (NRF, 2016 M3A7B4905619), and the National Research Foundation of Korea (NRF) grant (2018R1A2B2005205).

## Author contributions
J.L., D.-M.L., H.S.J., and S.M.K. conceived and designed the experiments. Y.J. and J.P. contributed to the synthesis of CNTFs using the direct spinning method. J.L., D.-M.L., H.S.L., and Y.-K.K. performed the CSA treatment experiments. J.L., H.S.J., and S.M.K. wrote the paper. C.R.P., H.S.J., and S.M.K. supervised the project. All the authors discussed the results and contributed to the preparation of the paper.

## Additional information

**Competing interests:** The authors declare no competing interests.

