## [Peer Review File · Nature Communications]

Reviewers' comments:

Reviewer #1 (Remarks to the Author):

In this manuscript, an ideal spinning method for fabricating highly dense and aligned carbon nanotube fiber is demonstrated by combining direct spinning and a simple/one step protonation through CSA treatment. This significantly improves the mechanical and electrical properties of the assembled carbon nanotube fibers. The effects of oxidation and different crystallinity on the alignment and properties improvement were systematically investigated for further parameter optimization of the CSA process. The ideal approach for direct spinning fibers and post CSA treatment is innovative and the scale-up potential is also demonstrated. As a result, the manuscript is suitable for publication in Nature Communications. It will be helpful if the authors can further clarify the correlation between alignment and the improved mechanical properties. For example, the oxidized CNF by heating and plasma treatment has lower alignment as compared to CNTF without oxidation, but display greater specific tensile strength. This is confusing. Additionally, different crystallinities of CNTF were tested, and it shows that the oxidation will actually improve the better property for low IG/ID. Will oxidation alter the crystallinity and IG/ID? How do different crystallinities impact the degree of oxidation and later DOP during CSA. A correlation between crystallinity after oxidation treatment with DOP and mechanical properties will be helpful to establish this trend.

Reviewer #2 (Remarks to the Author):

Ideal spinning method..... Jaegeum Lee et al.

Main claims of the paper and originality

It is that using the post spinning treatment process for direct spun CNT fibres, strengths in excess of 4 N/tex can be achieved. The importance of this value is that it puts the strength in the range of the very best traditional carbon fibre now available, while of course far exceeding the properties of the traditional fibre in respect of electrical conductivity and toughness in bending (knot efficiency).

The paper demonstrates that the post spinning process, stretching in CSA (chloro sulphonic acid) and then washing in water and acetone, increases the as-spun strength by somewhat over 100%.

This paper follows closely in the wake of another Korean paper: Cho et al. Carbon (2018) doi 10.1016/j.carbon.2018.04.071, It was accepted for publication on 24/04/18, and would have been on line from then but it is not quoted by the authors of this submission. It is from different Korean groups and I have to see it as prior art. The Carbon paper describes an almost identical post treatment in CSA with further fibre stretching and cites an improvement compared with the as-spun fibre of 130%. However, the key difference is that the as spun fibre used by the communication under review was much stronger to start with: 2.2 N/tex compared with 0.95. The authors describe this higher starting figure as a result of "their efforts to optimise the direct spinning process---" (p.5 of ms.) However, as this is a key ingredient which might justify their title of " Ideal spinning method...." , it should be described in some detail. The reference closest to it in the text does not help.

It appears that whatever has been done to modify the original spinning process to get 2.2N/tex shows that that improvement is ADDITIVE to the post spinning improvement of the CSA stretching treatment, whereas alternative post treatments such as thermal anneal or plasma treatments give improvements which are little enhanced by subsequent CSA stretching treatment. I believe this is the

essential new claim of the paper, but I have had to dig deeply into the ms to isolate it.

Scientific support for claims.

Given this claim, and indeed the importance of having achieved strengths as high as 4.4N/tex on a long gauge length sample, it is now a matter of deciding whether the science of the paper is sufficient to justify that claim. I have no issue with the quality of the experimental measurements, however the concern centres on what is not reported as well as some of the logic in the discussion.: -

1. It is not completely clear whether the values for strength and conductivity as shown in Figs 2 and 3 (a,b and g) are best values seen or average values. There is a hint on page 9 with the statement "as high as 4.4 N/tex" implying that the graphs are mean values. However, a mean of 4.08 implies a small standard deviation. If the graphs are average values then what was the population? Put it another way, how many separate synthesis and treatment runs were behind each of the points on Figs 4 (b and c)
2. The change in ratio of absolute/specific measurements Figs 2/3(a) and (g) should give an indication of the densification achieved by the process. However, Fig 3 shows a much greater densification for the electrical measurements than for the mechanical so the data could hardly be on the same fibre. Is this because there has been a measure of selectivity in choosing data?
3. The comparative temperature profile measurements in Fig 3 are at best qualitative, and no more than a suggestion of an improvement of thermal conductivity, even if it might be expected.
4. One would expect a reduction in knot efficiency with the enhancement in strength on CSA treatment. So why are we not told what the knot efficiency was for the as-spun fibre ?
5. There is no mention of whether the as-spun fibre had received any liquid treatment at all as a result of spinning. Perhaps spinning was done through a water bath for example.
6. The authors study the effect of thermal and plasma treatments prior to CSA treatment. Surely, they will have tried such treatments after the CSA stage. The result would be interesting, important perhaps. Why were these data not reported ? but if not yet carried out they should be.
7. In the first paragraph of the main text it is stated that the authors have demonstrated a continuous method for the post treatment (which as written is the essence of the ms.) Quick examination of Fig 4(d) might reinforce this view. However, closer reading of the submission suggest that the post treatments were never carried out on-line, and the establishment of these treatments on line is speculation. Also, the arguments that the thermal and plasma treatments cannot be put on-line are spurious.
8. A final point which raises a question regarding scientific quality, concerns the authors reference to the structural parameter given by the Raman I_g/I_d ratio as "crystallinity". Crystallinity is a well established scientific term and it is to do with three-dimensional periodic positional order. The I_g/I_d Raman ratio gives an indication of the purity and the perfection of the nanotubes with respect to all carbonaceous (sp^2) material in the sample. It is nothing to do with 3D periodic order. All references to "crystallinity" should be replaced with something like "purity and perfection".

Presentation

The English is rather good, however, there is far too much material to constitute a note. The Figures are nicely drawn but reproduce far too small to make their reading easy, particularly the micrographs

which are not needed in such quantities. These purport to show densification, but this is well established, and something which has been analysed in detail already (e.g Cho in the Carbon paper). The authors frequently refer to samples as CSA-heat-CNTF, whereas, the heat or plasma treatments come before the CSA treatments.

Rescue?

Can this submission be rescued as a Nature communication? It does contain a claim which is a significant advance in the field. However it would need to focus on this claim and eliminate much additional material. As it stands there is more than enough material here for a full paper which I expect the authors are contemplating anyway. There may be a case for a new review if the authors were to resubmit a much more focused note, explaining their improvement to the spinning process and demonstrating that CSA treatment still give a 100% strength improvement, even for the improved as-spun fibres.

Responses to Reviewers' Comments

Manuscript Title: Ideal spinning method for high-performance carbon nanotube fiber

Below are our responses to the reviewers' comments.

We thank the reviewers very much for their time and effort in providing us valuable comments on this manuscript. We will respond individually to the concerns and comments of the reviewers below.

(1) Answers to the comments raised by the reviewer #1

In this manuscript, an ideal spinning method for fabricating highly dense and aligned carbon nanotube fiber is demonstrated by combining direct spinning and a simple/one step protonation through CSA treatment. This significantly improves the mechanical and electrical properties of the assembled carbon nanotube fibers. The effects of oxidation and different crystallinity on the alignment and properties improvement were systematically investigated for further parameter optimization of the CSA process. The ideal approach for direct spinning fibers and post CSA treatment is innovative and the scale-up potential is also demonstrated. As a result, the manuscript is suitable for publication in Nature Communications.

Q1. It will be helpful if the authors can further clarify the correlation between alignment and the improved mechanical properties. For example, the oxidized CNF by heating and plasma treatment has lower alignment as compared to CNTF without oxidation, but display greater specific tensile strength. This is confusing.

A1. We thank the reviewer for the important critique. Before answering the question, we should notify that we had to remove the results about thermal and plasma treatment in the

revised manuscript according to the other reviewer's strong suggestion. In the revised manuscript, we spent the significant portion of the revised manuscript on the synthesis of high strength as-spun CNTFs and focused only on the effect of defect density of CNTs on the efficiency of CSA treatment, which turned out to be much more critical.

To answer the question, for CNTF with low I_G/I_D , even though the degree of alignment of oxidized CNTF after CSA treatment was lower than that of as-spun CNTF after CSA treatment, the specific tensile strength of oxidized CNTF after CSA treatment was higher than that of as-spun CNTF after CSA treatment, but the degree of densification of oxidized CNTF after CSA treatment was also higher. On the contrary, for CNTFs with medium and high I_G/I_D s, the evolution of the CNTF structures after CSA treatment had the same trend, but the specific tensile strengths of oxidized CNTFs after CSA treatment was lower than those of as-spun CNTFs after CSA densification. Therefore, we speculate that there are actually a lot of factors determining the final strengths of densified CNTFs such as the original internal structures and different interaction between CNT bundles as a result of different characteristics of the constituent CNTs. Thus, although the increase of alignment of the same CNTF sample after CSA treatment can be directly correlated to increase of final strength, it is difficult to correlate the degree of the alignment of different samples to their final strengths. We included the related discussion about this question in our revised manuscript as below.

Inserted sentences (from the 6th sentence to the 7th sentence of the 1st paragraph on page 12):

However, the final strengths of CSA-CNTFs can be affected by various factors such as the original internal structures and different interaction between CNT bundles as a result of different characteristics of the constituent CNTs, so these characterizations do not fully explain the different degree of improvement in CNTFs that had different defect densities. Based on our observation during CSA densification experiments that

the low-defect CNTFs swelled in CSA much faster than the other CNTFs did, the structural rearrangement from the original structures should be more active in the CNTFs that have low defect density than in CNTFs that have high defect density.

Q2. Additionally, different crystallinities of CNTF were tested, and it shows that the oxidation will actually improve the better property for low IG/ID. Will oxidation alter the crystallinity and IG/ID? How do different crystallinities impact the degree of oxidation and later DOP during CSA. A correlation between crystallinity after oxidation treatment with DOP and mechanical properties will be helpful to establish this trend.

A2. The efficiency of the CSA treatment in this work is dependent on the wettability of CNTs by CSA. It has been well established that the solvation of CNTs by CSA highly depends on (1) the defect density of CNTs (ACS Nano (2010) 4, 3969). Additionally, our group recently reported that (2) the oxidation state of CNTs also critically affects the solvation of CNTs by CSA (Small (2017) 13, 1707731). Thus, we originally intended to test the effect of defect density (represented by I_G/I_D) and oxidation on the efficiency of the CSA treatment. That is why we prepared as-spun CNTFs that had different I_G/I_D s and also included the oxidation step prior to CSA treatment.

Since the purpose of oxidation treatments performed in our work was not to alter the crystallinity but to incorporate small portion of oxygen onto CNTs, the oxidation conditions for thermal or plasma treatment were not harsh enough to alter the crystallinity (I_G/I_D). In our previous paper (Small (2017) 13, 1707731), we confirmed that the crystallinity of CNTs (I_G/I_D) did not alter even after the thermal treatment at 400 °C in air. In this study, we performed the thermal treatment at 300 °C in air or plasma treatment under Ar (20 slm) and O₂ (20 sccm) atmosphere at 250 W. Based on the characterization results as well as our

previous work (Small (2017) 13, 1707731), we speculated that the oxidation improved DOP due to the incorporation of oxygen onto CNTs and thus increased the density of CNTFs the most effectively in most cases. Again, since we removed the results for the densification of oxidized CNTFs in the revised manuscript, we could not add related discussion in the revised manuscript. We apologize the reviewer.

(2) Answers to the comments raised by the reviewer #2

Main claims of the paper and originality

It is that using the post spinning treatment process for direct spun CNT fibres, strengths in excess of 4 N/tex can be achieved. The importance of this value is that it puts the strength in the range of the very best traditional carbon fibre now available, while of course far exceeding the properties of the traditional fibre in respect of electrical conductivity and toughness in bending (knot efficiency).

The paper demonstrates that the post spinning process, stretching in CSA (chloro sulphonic acid) and then washing in water and acetone, increases the as-spun strength by somewhat over 100%.

This paper follows closely in the wake of another Korean paper: Cho et al. Carbon (2018) doi 10.1016/j.carbon.2018.04.071, It was accepted for publication on 24/04/18, and would have been on line from then but it is not quoted by the authors of this submission. It is from different Korean groups and I have to see it as prior art. The Carbon paper describes an almost identical post treatment in CSA with further fibre stretching and cites an improvement compared with the as-spun fibre of 130%. However, the key difference is that the as spun fibre used by the communication under review was much stronger to start with: 2.2 N/tex

compared with 0.95. The authors describe this higher starting figure as a result of “their efforts to optimise the direct spinning process---“ (p.5 of ms.) However, as this is a key ingredient which might justify their title of “ Ideal spinning method....” , it should be described in some detail. The reference closest to it in the text does not help.

It appears that whatever has been done to modify the original spinning process to get 2.2N/tex shows that that improvement is ADDITIVE to the post spinning improvement of the CSA stretching treatment, whereas alternative post treatments such as thermal anneal of plasma treatments give improvements which are little enhanced by subsequent CSA stretching treatment. I believe this is the essential new claim of the paper, but I have had to dig deeply into the ms to isolate it.

Authors' response

First of all, we really appreciate the reviewer for the insightful review. We thoroughly read the comments and prepared for the refocused and substantially revised manuscript following the reviewer's request.

We were aware of the Carbon paper by Cho *et al.* at the time of initial submission and cited it in the original manuscript. However, as you pointed out, we came to an agreement that the advance made in this paper was not fully highlighted in the original manuscript. In the original manuscript, we intended to differentiate this work from the previous report by Cho *et al.* by providing the comprehensive investigation into the effect of defect density and oxidation of CNTs on the CSA treatment. It is known that the mechanism by which CNTs dissolve in CSA is protonation, and the solubility of CNTs in CSA is determined by defect density (ACS Nano (2010) 4, 3969) and oxidation state of CNTs (Small (2017) 13, 1707731). Hence, we hypothesized that the degree of infiltration of CSA into CNTFs should be critically affected by the defect density and the oxidation state of constituent CNTs and thus

the two characteristics of constituent CNTs critically might affect the final strengths of CNTFs after CSA treatment. As expected, the two factors did affect the strength, but the defect density turned out to be much more critical. However, in all cases, the CSA treatment of oxidized CNTFs resulted in more densified CNTFs, but more densified structures of CNTFs did not always lead to higher specific tensile strengths of CNTFs. We speculated that the strength of CNTF after CSA treatment should be determined by the combination of various factors such as initial internal structures, alignment, density and degree of interaction between CNT bundles as a result of different characteristics of the constituent CNTs. Despite new knowledge we acquired in the study, we accept that the way presenting the results in the original manuscript may seem not quite appealing or clearly distinctive from the paper by Cho, *et al.* in terms of the utilized method, because prior oxidation of CNTF with high I_G/I_D did not lead to the highest mechanical property after CSA treatment.

In this respect, we appreciate the reviewer for the indication that the excellent mechanical property of our as-spun CNTF should be the point to be highly emphasized. Following the reviewer's suggestion, we prepared for a substantially refocused and revised manuscript. We spent a significant portion of the revised manuscript on describing the optimization process of CNTF synthesis, which enabled us to synthesize CNTFs dominantly composed of double-walled CNTs with controlled defect density while possessing among the best mechanical properties. Instead, we removed the results relating to thermal or plasma treatment from the original manuscript and only focused on the effect of defect density on the efficiency of the CSA treatment, which turned out to be much more critical. As a result, we strongly believe that the revised manuscript can, more clearly and effectively, deliver what we have learned in this study. We thank the reviewer very much again; we may consider a separate publication about the effect of oxidation on CSA treatment for comprehensive understanding of CSA densification mechanism.

Accordingly, the Figures were totally redesigned in the revised manuscript. In the schematic representation in Fig. 1, we removed thermal and plasma treatment steps. We included the results about synthesis of CNTFs in Fig. 2 and Table 1. The effect of CSA treatment on as-spun CNTFs having various defect density was shown in Fig. 3. In Fig. 4, the analysis of structural change of CNTFs with high I_G/I_D after CSA treatment was displayed. In Fig. 5, the properties of the best densified CNTFs were shown and compared with other high-performance materials, and continuous spinning method for highly densified and aligned CNTFs was proposed.

Scientific support for claims.

Given this claim, and indeed the importance of having achieved strengths as high as 4.4N/tex on a long gauge length sample, it is now a matter of deciding whether the science of the paper is sufficient to justify that claim. I have no issue with the quality of the experimental measurements, however the concern centres on what is not reported as well as some of the logic in the discussion.

Q1. It is not completely clear whether the values for strength and conductivity as shown in Figs 2 and 3 (a,b and g) are best values seen or average values. There is a hint on page 9 with the statement “as high as 4.4 N/tex” implying that the graphs are mean values. However, a mean of 4.08 implies a small standard deviation. If the graphs are average values then what was the population? Put it another way, how many separate synthesis and treatment runs were behind each of the points on Figs 4 (b and c)

A1. We are sorry for the unprofessional presentation of the measurement values. The values for strength in Figs 2 and 3 were average values. In the revised manuscript, for the

measurement of specific tensile strength, specific tensile modulus, and knot efficiency, we provided the standard deviations as well as the number of samples along with the average values. For the experiments to test the effect of the defect density, the CNTFs with each level of the defect density were obtained from one continuous synthesis run in order to rule out the inevitable run-to-run variations of the synthesis. We also removed a couple of meters of CNTFs from both ends. Since the main focus of this study was on the improvement of CNTF's mechanical property of CNTF, the specific electrical conductivity of each pristine and densified CNTF was measured from one sample due to the limited sample availability. Therefore, we placed the data for the change of the specific electrical conductivity after CSA treatment for CNTFs with various defect densities in the Supplementary Information, rather than in the main revised manuscript. Nevertheless, as our measurement was accurate, and we do not emphasize much about the specific electrical conductivity, we believe that there is no logical leap in our discussion regarding the specific electrical conductivity.

Inserted sentences (from the 1st sentence to the 3rd sentence of the 1st paragraph on page 14):

Compared to the improvement of the specific tensile strength, the specific electrical conductivity showed a marginal improvement after CSA treatment (Fig. S4), even though the specific electrical conductivity of our as-spun CNTFs is already comparable to that of metals (Fig. 5b). This lack of improvement implies that the specific electrical conductivity of CNTF cannot be improved above a certain point by only structural rearrangement of CNTFs, and may be restricted by the intrinsic properties of CNTs such as O/C or defect density (I_G/I_D). Therefore, the optimal internal structures as well as the characteristics of constituent CNTs for the best mechanical properties of CNTFs may be different from those for the best electrical properties of CNTFs.

Inserted Figure (Fig. S4 in Supplementary Information):

Fig. S4. Specific electrical conductivity of CNTFs having low, medium, and high I_G/I_D before and after CSA treatment.

Q2. The change in ratio of absolute/specific measurements Figs 2/3(a) and (g) should give an indication of the densification achieved by the process. However, Fig 3 shows a much greater densification for the electrical measurements than for the mechanical so the data could hardly be on the same fibre. Is this because there has been a measure of selectivity in choosing data?

A2. We appreciate the reviewer for this comment. Yes, we checked our data and confirmed that this discrepancy was caused from the selectivity in choosing data (there was the difference in cross-sectional areas between the samples with which we measured the tensile strength and electrical conductivity). In the case of direct spinning, there are many cases where cross-section of as-spun CNTF is not uniform along its axis even though its linear density is uniform along its axis. Since the synthesis reaches a steady state during the continuous direct spinning of CNTF, the linear density is likely to be kept constant. However,

the packing of the as-spun CNTF by water during the synthesis entails a random characteristic. In this study, we analyzed the cross-section of each type of CNTF for two to four times and realized that there is quite a variation in cross-sectional area within a same sample (Fig. R1). On the contrary, the linear density is not only uniform within a continuous fiber, but also easy to measure by vibroscopic method with the use of FAVIMAT. Besides, the measured linear density was highly consistent within a sample. Also, our group has published the paper concerning how to accurately measure the linear density of CNTFs by the vibroscopic method (RSC Advances (2017) 7, 8575).

Fig. R1. Cross-sectional SEM images of as-spun CNTFs having high I_G/I_D used for the measurement of (a) tensile strength and (b) electrical conductivity, respectively. The cross-sectional area of the CNTF in (a) is $5.5 \times 10 \mu\text{m}^2$ and that in (b) is $7.0 \times 10 \mu\text{m}^2$.

Accepting this comment seriously, we decided not to report all the properties of CNTFs requiring the measurement of “cross-sectional area”, such as density, tensile strength, tensile modulus, and electrical conductivity. Instead, we decided to report only properties of CNTFs requiring the measurement of the linear density, such as specific tensile strength, specific tensile modulus, and specific electrical conductivity.

Q3. The comparative temperature profile measurements in Fig 3 are at best qualitative, and no more than a suggestion of an improvement of thermal conductivity, even if it might be expected.

A3. We have been optimizing the apparatus for the accurate measurement of the thermal conductivity of CNTFs based on self-heating method (Science (2015) 34, 1083, Nano Lett. (2012) 12, 4848, Adv. Mater. (2014) 26, 4521). A lot of efforts such as use of high-resolution micro IR camera (C10614-02, Hamamatsu Photonics, Japan) for CNTFs with very small diameters and a series of calibrations have been made to minimize experimental uncertainty (Fig. R2). However, the optimization is still on going.

Therefore, to avoid any uncertainty or confusion of the potential readers, we decided to remove the related thermal images showing comparative heat transport in the revised manuscript. We will be soon able to report the quantitatively accurate thermal conductivity of the CNTFs.

Fig. R2. Thermal conductivity measurement experimental setup for CNTF at KIST. A series of optimization and calibration is still on going.

Q4. One would expect a reduction in knot efficiency with the enhancement in strength on CSA treatment. So why are we not told what the knot efficiency was for the as-spun fibre?

A4. The knot efficiency of the as spun CNTF was measured and included in Fig. 5a. As you expected, there was reduction in the knot efficiency with the enhancement in strength by CSA treatment. This was discussed in the revised manuscript as below.

Inserted sentences (from the 1st sentence to the 2nd sentence of the 2nd paragraph on page 13)

Our CSA-CNTFs have uniquely high flexibility, which is well represented by the high knot efficiency of $48 \pm 15\%$ ($n = 5$) (Fig. 5a). This value is lower than the knot efficiency of as-spun CNTF ($67 \pm 9\%$, $n = 7$), but higher than that of high-strength carbon fibers²⁹.

Inserted Figure (Fig. 5a)

Fig. 5(a) Normalized specific stress-strain curves of CNTF, knotted CNTF, CSA-CNTF, and knotted CSA-CNTF.

Q5. There is no mention of whether the as-spun fibre had received any liquid treatment at all as a result of spinning. Perhaps spinning was done through a water bath for example.

A5. The as-spun CNTFs passed through a water bath, which also plays a role in sealing the reactor, prior to any characterization or treatment. This was already mentioned in the experimental section in the Supplementary Information. In the revised Supplementary Information, this can be found in the 6th sentence of the 1st paragraph on page 2; *Aerogel-like CNT assembly formed in vertical furnace was transformed into a fiber passing through a water bath.*

Q6. The authors study the effect of thermal and plasma treatments prior to CSA treatment. Surely, they will have tried such treatments after the CSA stage. The result would be interesting, important perhaps. Why were these data not reported? but if not yet carried out they should be.

A6. The purpose of thermal and plasma treatment prior to CSA treatment is not to improve the mechanical or electrical property of CNTFs, but to test the effect of oxidation of CNTs on the efficiency of CSA treatment. As mentioned in our response to “Main claims of the paper and originality”, the solubility of CNTs in CSA is dependent on the defect density (ACS Nano (2010) 4, 3969) and oxidation state of CNTs (Small (2017) 13, 1707731). This was also stated in the original manuscript (the 6th sentence of the 1st paragraph on page 4); *Crystallinity and oxidation state of CNTs significantly affect the solubility of CNTs in CSA^{5,17}, and therefore the densification efficiency, so these traits were investigated.* Thus, by thermal and plasma treatments prior to CSA, we intended to test the effect of oxidation on the densification efficiency by CSA treatment and thus final strength CNTFs after CSA treatment. Therefore, there was no reason for us to include thermal and plasma treatments of CNTFs after CSA treatment in this study. Anyway, since we decided to refocus and substantially revise the original manuscript following the reviewer’s request, we removed the results for thermal and plasma treatments from the original manuscript.

Q7. In the first paragraph of the main text it is stated that the authors have demonstrated a continuous method for the post treatment (which as written is the essence of the ms.) Quick examination of Fig 4(d) might reinforce this view. However, closer reading of the submission suggest that the post treatments were never carried out on-line, and the establishment of these treatments on line is speculation. Also, the arguments that the thermal and plasma treatments cannot be put on-line are spurious.

A7. We apologize the reviewer for causing such a confusion. In the first paragraph of the main text, we stated that we demonstrate the “feasibility” of highly efficient and entirely continuous fiber-spinning method from the synthesis of carbon nanotubes (CNTs) to the

fabrication of highly densified and aligned carbon nanotube fibers (CNTFs) within one-minute processing time. We did not intend to claim that we actually have tested the on-line continuous process. The optimum continuous process we proposed in the original manuscript excluded the thermal or plasma treatment, because the best CNTF resulted from the densification of CNTF with high I_G/I_D without prior oxidation. For this process, we strongly believe that the proposed continuous process is feasible because the time scale of each unit process is < 1 min. The only reason we did not demonstrate continuous process was the usage of superacid CSA. However, if there is appropriate safety equipment installed, we cannot find any reason that the densification process cannot be performed on-line, because on-line stretching process of conventional fibers is well utilized in fiber industry. Regarding the comment on thermal and plasma treatments, we originally intended to include the thermal treatment to examine the effect of oxidation and the plasma treatment to reduce the oxidation time (30 min for thermal and 5 min for plasma) to check the feasibility of inclusion of oxidation step as a continuous process. Since we utilized the atmospheric plasma treatment, which has been used as a continuous process for the stabilization of carbon fiber at KIST here, we thought that the plasma treatment could be added as a continuous process. We agree with the reviewer that the thermal treatment is hardly performed in a continuous manner. We are sorry for not having a clear indication for the purpose of plasma treatment in the original manuscript. In the refocused and revised manuscript, the results for thermal and plasma treatments were removed anyway. Accordingly, we revised the description about the proposed process in a much simpler way as follows.

Previous sentences (from the 4th sentence to the of the 10th sentence of the 2nd paragraph on page 11):

The proposed method has the additional advantage of potential scalability; the CNTFs can be synthesized in a brief (< 1 min) continuous process. Considering the

properties of various CNTFs obtained in this work (Fig. 4b and c) and the simplicity of process, the optimal process is to synthesize a CNTF that has high I_G/I_D , then to pass it directly through a CSA bath and an acetone coagulation bath consecutively under appropriate tension (Fig. 4d). Synthesis of CNTFs with high I_G/I_D is always desirable, because high I_G/I_D provides large room for further improvement of properties by the CSA treatment, although the properties of as-spun CNTF may be similar regardless of I_G/I_D . When high I_G/I_D is not achievable, use of an oxidation step before CSA treatment might help maximize the effect of CSA treatment. The inclusion of the oxidation step has the additional advantage of purifying the samples. Thermogravimetric analysis reveals that CNTFs subjected to CSA treatment after the oxidation step contain significantly less Fe residue than do CNTF without oxidation (Fig. S8). Thus, depending on the purpose and properties of as-spun CNTF, one can design an optimal continuous process.

Revised sentences (the 2nd paragraph on page 14):

The proposed method has the additional advantage of potential scalability. The time scale of each unit process is < 1 min, so the CNTFs can be synthesized in a brief continuous process (Fig. 5c). We have confirmed that the degree of improvement largely depends on the defect density of CNTs; therefore, the key to the successful process lies in the synthesis of high-performance CNTFs that are composed of low-defect CNTs.

Q8. A final point which raises a question regarding scientific quality, concerns the authors reference to the structural parameter given by the Raman I_G/I_D ratio as “crystallinity”. Crystallinity is a well established scientific term and it is to do with three-dimensional periodic positional order. The I_G/I_D Raman ratio gives an indication of the purity and the

perfection of the nanotubes with respect to all carbonaceous (sp²) material in the sample. It is nothing to do with 3D periodic order. All references to “crystallinity” should be replaced with something like “purity and perfection”.

A8. We are sorry for the improper use of terminology. To replace “crystallinity”, we carefully checked the relevant series of works on the dissolution of CNTs by CSA by Prof. Matteo Pasquali from Rice University (USA), because most of publications related to CSA treatment with CNTs came from his group. They used the term “defect density” to represent the structural perfection of CNTs and assessed defect density by I_G/I_D obtained from Raman spectroscopy (ACS Nano (2010) 4, 3969-3978). Thus, we replaced all references to “crystallinity” with “defect density” in our revised manuscript. We thank the reviewer very much for this indication.

Presentation

Q1. The English is rather good, however, there is far too much material to constitute a note.

A1. Following the reviewer’s request, we removed a substantial portion of our results especially for thermal and plasma treatments. Instead, we more emphasized the synthesis of as-spun CNTFs and the effect of the defect density on the efficiency of CSA treatment and thus the final strength of CNTF after CSA densification. As a result, we now feel that the revised manuscript is more concise and highly focused. We think now that the material is not too much to constitute a note.

Q2. The Figures are nicely drawn but reproduce far too small to make their reading easy, particularly the micrographs which are not needed in such quantities. These purport to show

densification, but this is well established, and something which has been analysed in detail already (e.g Cho in the Carbon paper).

A2. In the revised manuscript, Figures from 2 to 5 are newly drawn or relocated. Overall, we improved the readability of the figures and increased the size of font, especially in Figure 5b. We left the cross-sectional SEM images of CNTF with high I_G/I_D before and after the treatment in the revised manuscript (Fig. 4a-d), while those of the other samples are displayed in the Supplementary Information.

Q3. The authors frequently refer to samples as CSA-heat-CNTF, whereas, the heat or plasma treatments come before the CSA treatments.

A3. We are sorry for the confusion. The results about thermal treatment and plasma treatment were removed in the revised manuscript, so we didn't need to refer to these samples.

Rescue?

Can this submission be rescued as a Nature communication? It does contain a claim which is a significant advance in the field. However it would need to focus on this claim and eliminate much additional material. As it stands there is more than enough material here for a full paper which I expect the authors are contemplating anyway. There may be a case for a new review if the authors were to resubmit a much more focused note, explaining their improvement to the spinning process and demonstrating that CSA treatment still give a 100% strength improvement, even for the improved as-spun fibres.

Authors' response

We really appreciate the reviewer for mentioning the significance of this work and thus providing us a precious opportunity for rescue. Fully following the reviewer's suggestion, we have prepared for a much more focused manuscript. In the revised manuscript, we shed light on the improvement of the direct spinning process of as-spun CNTFs by providing enough results and discussion (See Table 1 and Fig. 2 in the revised manuscript). In addition, we highlighted that CSA treatment still can give a 100% strength improvement even for the improved as-spun CNTFs having a specific tensile strength higher than 2 N/tex as below.

Inserted sentences (from the 5th sentence to the 8th sentence of the 1st paragraph on page 11):

In general, an effect of a post-treatment is likely to be amplified when the properties of as-spun CNTFs are relatively low, because in these CNTFs, the room for the improvement is large. When the synthesis is well optimized, so the as-spun CNTFs are of high quality as in this work, efficiency of a treatment may not be so obvious. Nevertheless, the CSA treatment nearly doubled specific tensile strength and nearly quadrupled specific tensile modulus (from 48.3 ± 7.4 N/tex to 187.5 ± 7.4 N/tex) of CNTFs that had high I_G/I_D . The results indicate that the CSA densification can effectively maximize the mechanical properties of CNTFs as long as as-spun CNTF has a low defect density.

We are so grateful to the reviewers for taking the time to provide us with this review. We feel that the revised version of this manuscript is much better suited for publication in the *Nature Communications* following this review.

Reviewers' comments:

Reviewer #1 (Remarks to the Author):

The manuscript was completely rewritten as compared with the previous version. The effects of oxidation and plasma treatment on the CNTF alignments and mechanical properties were removed, in which some inconsistent data were identified previously. I expected to see more systematic works on the control of alignment on mechanical and electrical properties, which are critical for carbon fibers and carbon nanotube fibers. The current version focuses on the effects of defect density on the mechanical properties. A substantial part of the revised manuscript included the fiber process and how to control the defect density. This part is not essential for the manuscript and should be placed in the supplementary information. It is well understood that the defect density will significantly impact the strength of fiber materials in the field.

The innovation on the CSA process and how defects manipulating the crystallinity (even the authors removed this term), compactness, and microstructure alignment and orientation were not well explained. The scientific understanding will be important and only limited data show the difference (pore and density) without detailed investigation.

Overall, the mechanical properties are still very impressive, and fundamental understanding is not there yet despite that the defect density was highlighted as the dominated factor. The interplay between defect control and CSA process is not well explained. To justify for the publication in Nature Communication, the authors need to show the difference of innovation as compared with previous Carbon paper by a different Korean group if CSA process was already demonstrated, or a significant advancement in the fundamental understanding of the CSA is required.

Reviewer #2 (Remarks to the Author):

Referee's comments on the revised m/s by page:

2 I. 22 remove 'entirely' and substitute 'potentially'. The authors have not demonstrated a continuous process embodying the CSA step.

5. I. 91 insert new sentences possibly using this place to convey other information about the process which does not appear here or in ~SOM. One sentence should mention the nature of the ceramic used in the reactor tube and its diameter, the other should indicate how the reactor volume was sealed to permit fibre egress. Somewhere else it was hinted that the fibre was drawn out through a water bath. The significance is that the water, if indeed that is what was used, also condenses the fibre, and must be the reason why the directly spun fibre showed the degree of condensation seen in Fig 4. There are other places this information could appear, but it should be in the paper proper, not the SOM.

6. I. 106. How does ferrocene work in a bubbler, as it sublimes rather readily?
Ls. 109-112 I cannot understand this sentence, or even guess at what the authors are trying to convey. Could they please rewrite it, or split it into two or three sentences perhaps?

9. I. 153 The paper lacks a proper discussion of the relation of the Raman I_g/I_d ratio to 'defects', and it is suggested that it should be inserted here. I think it is generally held that I_d comes from regions close to edge termination of the graphitic layers. There are defects possible in CNTs, such as 5-7 pairs which may not leave a significant imprint on I_d , yet in kinking the tube may severely compromise its ability to be well aligned. On the other hand graphitic particles, possibly associated with 'blind' iron particles, but not belonging to the nanotubes which are being aligned may contribute strongly to I_d . I do not think that these comments invalidate the ratio measurements, but there does need to be a discussion of the meaning of this ratio,

with the added comment that it will not be affected by lack of perfect alignment of the CNTs.

11. I. 192 it would add to the impact of the paper, if, after the value for the specific modulus for T1000G, one for specific strength was also added.

I. 193 The paper requires another significant addition, and this might be the best place. We are invited to note qualitatively that from Figs 4a and c that the cross sectional area is reduced on CSA treatment, even though as mentioned earlier the tex only changed negligibly. Firstly the Figs 4 a-d do not reproduce well enough to be convincing in support of the text. I would strongly suggest that they are printed considerably larger. However, more than so, the paper here is crying out for some numbers as to the changes in volumetric density on CSA treatment. In the SOM the authors describe in some detail how they measured the fibre cross sectional area, so at the very least they should divide the actual values of the tex measurements by the measured cross section areas to give values for the volumetric density which can then be averaged over all the samples. However, this is not all. After CSA treatment the fibre is put into acetone, now acetone is a most effective condensation agent in its own right, so the question raises its head as to how much of the reduction in area is due to the CSA stage and how much due to the acetone. One might guess that it is largely due to the CSA stretch stage, but good science demands that it is checked. The authors should measure cross sectional areas both before and after an acetone treatment but without the CSA stage.

13 I.227 Fig 4 (see above)

14 I. 251. 'Unprecedentedly' and 'unique' are tautologous, and either word is a little OTT. I would suggest both words are substituted by just 'special'.

15. I. 259 Fig 5. Even at a fibre production rate of 15 m/min (eventual scale up may lead to speeds 10x faster !) an immersion of 1 min would require a bath 15m long. Fig 5 is misleading in this respect. I suggest that both baths are split in two along their vertical bisector, with jagged edges to remind the reader that the actual baths needed will be somewhat longer.

Responses to Reviewers' Comments

Manuscript Title: Ideal spinning method for high-performance carbon nanotube fiber

Below are our responses to the reviewers' comments.

We thank the reviewers very much for their time and effort in providing us valuable comments on this manuscript. We will respond individually to the concerns and comments of the reviewers below.

(1) Answers to the comments raised by the reviewer #1

The reviewer's comment.

The manuscript was completely rewritten as compared with the previous version. The effects of oxidation and plasma treatment on the CNTF alignments and mechanical properties were removed, in which some inconsistent data were identified previously. I expected to see more systematic works on the control of alignment on mechanical and electrical properties, which are critical for carbon fibers and carbon nanotube fibers. The current version focuses on the effects of defect density on the mechanical properties. A substantial part of the revised manuscript included the fiber process and how to control the defect density. This part is not essential for the manuscript and should be placed in the supplementary information. It is well understood that the defect density will significantly impact the strength of fiber materials in the field.

The innovation on the CSA process and how defects manipulating the crystallinity (even the authors removed this term), compactness, and microstructure alignment and orientation were not well explained. The scientific understanding will be important and only limited data show the difference (pore and density) without detailed investigation.

Overall, the mechanical properties are still very impressive, and fundamental understanding is not there yet despite that the defect density was highlighted as the dominated factor. The interplay between defect control and CSA process is not well explained. To justify for the publication in Nature Communication, the authors need to show the difference of innovation as compared with previous Carbon paper by a different Korean group if CSA process was already demonstrated, or a significant advancement in the fundamental understanding of the CSA is required.

Authors' response.

First of all, the authors feel sorry for the first reviewer in that we failed to address the reviewer's comments satisfactorily. Since we tried to do our best in following the strong suggestions of the other reviewer, we had no choice but to refocus the manuscript and remove substantial portion of the results that had been included in the original submission.

However, in the second revision process, actively accepting the reviewer's comment, we largely revised the previous version of manuscript to present a comprehensive study on the CSA densification process and clearly show the significance of the degree of protonation (DOP) on the improvement of the mechanical properties of CNTFs during CSA densification process. To provide fundamental understanding on the CSA densification process, we newly performed additional series of experiments for the systematic study on the process optimization and reorganized the oxidation results in the initial submitted version for emphasizing the significance of the proper DOP on the efficiency of CSA densification process. Therefore, in the revised manuscript, we newly added two sections with two extra figure sets (Figs. 3 and 6). Also, for smooth transition between the existing and newly added contents, we minorly modified several existing sentences as well. All changes were highlighted in red in the revised manuscript.

One section presented with Fig. 3 is about the optimization of CSA densification process. We considered that the stretching ratio and the immersion time are two most important experimental parameters in the CSA treatment. We performed new series of experiments to test the effect of stretching ratio and immersion time. In these series of experiments, how each parameter affects the structural rearrangement (i.e. alignment and density) of CNTF was independently analyzed and the change of mechanical strength was interpreted with the change of alignment and density. Here, we also suggest that controlling the degree of protonation (DOP) is the key to the process optimization.

Inserted section:

Optimization of CSA densification process. Starting with high-strength as-spun CNTFs that had medium I_G/I_D , we optimized the CSA densification process (Supplementary Information). Hereafter, CNTFs after the CSA densification process are referred to as “CSA-CNTFs”. First, we examined the effect of the stretching ratio (the change in length divided by the original length and multiplied by 100) of the CNTFs on the specific strength while they were immersed in CSA for 1 min. As stretching ratio increased from 0% to 10%, the specific tensile strength steadily increased (Fig. 3a), but at stretching ratio higher than 10%, most of the CNTFs broke. Stretching ratio of 10% resulted in the highest improvement in specific tensile strength, without breaking the CNTFs. We also optimized the immersion time. The specific tensile strength of CNTFs increased as the immersion time increased from 0.5 min to 1 min with the stretching ratio of 10%, but then decreased as the immersion time increased to 3 min (Fig. 3d).

The improvement in the specific strength of CNTFs depending on the stretching ratio of CNTFs during CSA densification process is correlated with internal structural changes of CNTFs: the alignment of CNTs along the fiber axis represented by

$I_{G\parallel} / I_{G\perp}$ (Fig. 3b) and the volumetric density (Fig. 3c). For appropriate comparison, all real values (Fig. S4) of specific strength, $I_{G\parallel} / I_{G\perp}$, and density were normalized to those of as-spun CNTF (Fig. 3). The method to measure the volumetric density is also described in the Supplementary Information. At 0% stretching ratio, the specific strength and the density of CSA-CNTFs increased, but the alignment was slightly degraded; this result means that the specific strength is mainly increased by the increased density. As the stretching ratio increased, the specific strength and alignment increased monotonously, but the density did not change much; indicating that the specific strength was mainly affected by the improvement in the alignment (Fig. 3a-c). Therefore, the improvement in the specific strength of CSA-CNTFs can be explained by the interplay between the alignment and the density.

The existence of an optimal immersion time can also be explained by the change in alignment and density (Fig. 3d-f). Basically, a certain DOP is required for CSA to penetrate the CNTFs and to solvate CNTs by protonation. Alignment and density both increased as immersion time increased from 0.5 min to 1 min; this result indicates that 0.5 min is too brief to cause sufficient structural rearrangement. However, excessive immersion time causes excessing DOP, so well-packed CNT bundles in the CNTF might disassemble. When the immersion time was further extended from 1 min to 3 min, the alignment did not increase and the density even decreased, causing overall decrease in the specific tensile strength of the CSA-CNTF. Thus, excessive immersion time is not helpful, and even might degrade, the properties of the CSA-CNTFs.

Inserted figure (Fig. 3):

Fig. 3. Optimization of CSA densification process. Evolutions in (a) normalized specific tensile strength, (b) normalized $I_{G\parallel} / I_{G\perp}$, and (c) normalized density of CNTFs having the medium I_G/I_D during CSA densification process with various stretching ratios (immersion time, 1 min). Evolutions in (d) normalized specific tensile strength, (e) normalized alignment, and (f) normalized density of CNTFs with various immersion times (stretching ratio 10%). All values are normalized to those of as-spun CNTFs.

Inserted figure (Fig. S4):

Fig. S4. Optimization of CSA densification process. Evolutions in (a) specific tensile strength, (b) $I_{G\parallel} / I_{G\perp}$, and (c) density of CNTFs having the medium I_G/I_D during CSA densification process with various stretching ratios (immersion time, 1 min). Evolutions in (d) specific tensile strength, (e) alignment, and (f) density of CNTFs with various immersion times (stretching ratio 10%). These are the real values of the data shown in Fig. 3.

The other section presented with Fig. 6 deals with the combinational effect of I_G/I_D and oxidation on the efficiency of CSA treatment. The results about the CSA treatment of oxidized CNTFs had previously been removed during the first revision, but in this revision, we included the results again to provide a deeper understanding on the CSA treatment process and emphasize the effect of DOP on the mechanical properties of CNTFs after CSA densification. Let us discuss about the importance of this section. The assumption that lies behind the current work as well as the work by Cho et al. in Carbon (2018) is that the breakage mechanism of CNTF is the slippage between CNTs, not the breakage of CNTs themselves, and therefore, the mechanical properties of individual CNTs are not fully transferred to the CNTF. This is because of loose structural network observed in most of the

previously reported CNTFs. It is generally accepted that structural arrangement of CNTs, not the properties of individual CNTs, determines the mechanical properties of the CNTF. The properties of individual CNTs would be fully reflected to the overall properties of CNTF only when CNTs are ideally aligned and packed in the CNTF. The focus of this paper is to induce the structural rearrangement of CNTs within a CNTF toward ideally aligned and packed structural organization, which occurs during the series of CSA treatment process: swelling of CNTs during immersion in CSA, aligning during stretching, and densification during coagulation process in acetone. Hence, the efficiency of our CSA treatment would be mainly determined by how well the CNTs in the CNTF are solvated by CSA. It has been well established that the solvation mechanism of CNTs by CSA is side wall protonation. So far, two factors have been revealed to affect the solubility of CNTs in CSA: defect density of CNTs and oxidation state. The importance of this newly added section is that we systematically investigated the effect of both factors on the efficiency of CSA treatment, which is one of the distinct originalities of this work from the work by Cho et al. in Carbon.

Inserted section:

Significance of DOP on improvement of the mechanical properties of CSA-CNTFs.

To increase understanding of the CSA densification process and the significance of DOP on the improvement of the mechanical properties of CSA-CNTFs, we also investigated the effect of oxidation on CSA treatment efficiency, by using CNTFs with low I_G/I_D (≈ 5) and CNTFs with high I_G/I_D (≈ 17). Although defect density is the main factor determining the solubility of CNTs in CSA^{5,25,26}, it was recently revealed that the oxidation of CNTs dramatically increases their solubility in CSA, because the oxygen-containing groups can serve as centers for protonation¹⁷. Our hypothesis is that controlling the DOP is important to achieve highest efficiency of the CSA treatment; if we are correct, the effect of the oxidation on the efficiency of the CSA

treatment would differ depending on the defect density in the CNTFs. Oxidation was conducted by heat treatment in air at 300 °C for 30 min. We first confirmed that the heat treatment in air did not significantly alter the defect density (Fig. S8) and that the ratio of oxygen to carbon (O/C) after the heat treatment in air did not vary significantly among CNTFs with different I_G/I_D (Fig. S9).

When the I_G/I_D was low, the CNTFs that had been heat-treated before CSA treatment (hereafter “heat-CSA-CNTFs”) had higher specific tensile strength than CSA-CNTFs (Fig. 6a). In this case, the oxidation helped the CNTFs with low I_G/I_D to have appropriate DOP, so the specific tensile strength of heat-CSA-CNTFs increased. As discussed in Fig. 3, the improvement of the mechanical properties after the CSA densification process can be explained by the interplay between the alignment and density improvement. When I_G/I_D was low, the alignment of heat-CSA-CNTF was slightly lower than that of CSA-CNTF (Fig. 6b), but heat-CSA-CNTF had much higher density than CSA-CNTF (Fig. 6c); this difference seems to compensate for the slightly lower alignment of heat-CSA-CNTF and results in overall improvement of specific tensile strength. On the contrary, when the I_G/I_D was high, heat-CSA-CNTFs had lower specific tensile strength than CSA-CNTFs (Fig. 6a). In this case, the 1-min immersion in CSA would cause DOP to be too high; this condition is not desirable to achieve high specific tensile strength as discussed in Fig. 3d-f. The alignment of CSA-CNTF is much higher than that of heat-CSA-CNTF (Fig. 6b). Also, the density of CSA-CNTF is already much higher than that of as-spun CNTF but heat-CSA-CNTF had a minor improvement in density compared to CSA-CNTF (Fig. 6c), so heat-CSA-CNTF had lower specific tensile strength than CSA-CNTF.

To confirm the hypothesis and compare the result with that in Fig. 3d-f, we performed CSA treatment with short immersion time (0.5 min) using heat-treated CNTF that had

medium I_G/I_D (Fig. 6d). The short immersion time yielded CNTF with increased strength; this result indicates that the optimal immersion time for heat-treated medium- I_G/I_D CNTF was shorter than 1 min, which was the optimal immersion time for medium- I_G/I_D CNTF in Fig. 3d-f. This result, combined with the result in Fig. 3d-f, suggests that the optimal immersion time that yields optimal DOP is dependent on the properties of the CNTs that comprise the CNTF; when I_G/I_D is relatively low, 1-min immersion is insufficient for the as-spun CNTF, but appropriate for the heat-treated CNTF, whereas when I_G/I_D is relatively high, 1-min immersion is appropriate for the as-spun CNTF, but excessive for heat-treated CNTF.

Inserted figure (Fig. 6):

Fig. 6. Combinational effect of I_G/I_D and heat treatment on the efficiency of CSA densification process. (a) Normalized specific tensile strength, (b) normalized $I_{G\parallel} / I_{G\perp}$, and (c) normalized density of as-spun CNTF, CSA-CNTF, and heat-CSA-CNTF with low and high I_G/I_D s. The immersion time and the stretching ratio in (a)-(c) were 1 min and 10%,

respectively. (d) Normalized specific tensile strength of heat-CSA-CNTF that had medium I_G/I_D as a function of immersion time. The stretching ratio in (d) was 10%. All real values (Fig. S10) of specific strength, $I_{G\parallel}/I_{G\perp}$, and density were normalized to those of as-spun CNTF.

Inserted figure (Fig. S10):

Fig. S10. Combinational effect of I_G/I_D and heat treatment on the efficiency of CSA densification process. (a) Specific tensile strength, (b) $I_{G\parallel}/I_{G\perp}$, and (c) density of as-spun CNTF, CSA-CNTF, and heat-CSA-CNTF with low and high I_G/I_D s. The immersion time and the stretching ratio in (a)-(c) were 1 min and 10%, respectively. (d) Specific tensile strength of heat-CSA-CNTF that had medium I_G/I_D as a function of immersion time. The stretching ratio in (d) was 10%. These are the real values of the data shown in Fig. 6.

In the previous revision, we included the detailed results about synthesis of high-strength

CNTFs by direct spinning method according to the other reviewer's request. Although the reviewer #1 recommended that it should be placed in the supplementary information, after long consideration, we decided to keep this part in the second revised manuscript, for two main reasons. The first reason is that our as-spun synthesized CNTF possess the best specific strength reported so far with sufficiently long gauge length (2 cm). Since 2014, there are only two papers that report the synthesis of high-strength CNTFs by direct spinning method, and to the best of our knowledge, there are no report on the synthesis of high-strength as-spun CNTFs exceeding 2 N/tex (So far, the highest specific tensile strength with sufficiently long gauge length was reported to be 2 N/tex (average value: ~1.6 N/tex) in Faraday Discussions (2014) 173, 47-65. More recently, the paper in ACS Nano (2015) 9, 7392-7398 reported the synthesis of CNTF with 1 N/tex. Thus, it would be important and meaningful to report our synthesis of high-strength as-spun CNTFs in the CNT synthesis community. The second reason is its own necessity in this paper. The reason why we controlled defect density is not to re-demonstrate the well-known fact that the defect density will significantly impact the strength of fiber, but to control the DOP as the defect density determines the solubility of CNTs in CSA. Although I_G/I_D is obviously an important parameter in CSA treatment, it would be difficult for other groups to perform this series of experiments because the synthesis of CNTFs with controlled I_G/I_D while keeping the tensile strength would require high level of synthesis knowledge and experience (Note that the mechanical properties of our as-spun CNTFs with different defect densities are not only superb but also very consistent).

The reviewer also asked us to show the difference of innovation of this manuscript as compared with the Carbon paper by Cho et al. While the Carbon paper is a simple demonstration showing the potential of wet stretching of CNTFs using CSA, our study is a

comprehensive study that ultimately aims at a continuous spinning process to produce ultra-high strength CNTFs. We listed below “originalities” as well as “advancements” of our work as compared with the Carbon paper by Cho et al.

Originalities

1. Fundamental approach to control the CSA treatment process based on the solvation mechanism of CNTs by CSA.

The most outstanding originality of this paper is that the paper presents a fundamental approach to control the CSA treatment process, rationally based on the mechanism by which CSA solvates CNTs. It has been well established that the solvation mechanism of CNTs by CSA is side wall protonation. In CSA, a super acid, the amphoteric nature of CNTs suggests that they will behave as a weak base and be protonated, directly resulting in the delocalization of positive charge over the entire nanotube. The electrostatic repulsion between protonated CNTs overcomes the van der Waals interaction between CNTs and causes solvation. It is also known that the solubility of CNTs in CSA is determined by the defect density of CNTs, typically related to I_G/I_D . In this study, we successfully synthesized the high-strength CNTFs that have nearly identical specific tensile strength but different defect densities. Using these CNTFs with different defect densities, we demonstrated that the CSA treatment is most effective to the CNTFs with low defect density (high I_G/I_D).

2. Comprehensiveness: combinational effect of I_G/I_D and oxidation on the efficiency of CSA treatment.

In addition to defect density, the oxidation of CNTs also affects their solubility in CSA as the oxygen-containing groups can serve as specific center for protonation. To make this study more comprehensive, we tried to investigate the combinational effect of defect density and

oxidation on the efficiency of CSA treatment. We also proposed that control of degree of protonation (DOP) of CNTs during the CSA treatment is the key to the improvement of mechanical and electrical properties.

3. Thorough analysis of change of structural rearrangement

In the Carbon paper by Cho et al. only the evidence was TEM images that show decreased voids. On the contrary, in this study, we analyzed the structural change more thoroughly and interpreted the change of mechanical properties in the framework of the structural rearrangement. For all the samples, we analyzed the alignment and density. The alignment of a CNTF was assessed by $I_{G\parallel} / I_{G\perp}$ using polarized Raman spectroscopy. The density of a CNTF was measured by dividing the linear density by the cross-sectional area. The accurate measurement of cross-sectional area is very important for CNTFs synthesized by direct spinning method because usually the geometry is far from cylindrical fiber. To accurately measure the cross-sectional area, we cut the cross-section of CNTFs using focused ion beam.

In addition, we demonstrated that the structural rearrangement is reflected to the actual mechanical properties. When the stress-strain curve is converted into modulus-strain curve, a clear difference between as-spun CNTF and highly aligned and well-packed CSA-CNTFs is revealed.

4. Combining wet-spinning technique for coagulation

There is a fundamental difference in the densification mechanism. In Cho's Carbon paper, the main driving force for densification after CSA treatment is capillary force by evaporating the solvent, which is a frequently used method as a post treatment to densify direct-spun CNTFs by infiltrating good solvents including NMP into the CNTFs. In this study, on contrary, we used poor solvent (acetone) to densify CNTF, which is well-established method as a coagulation process in wet-spinning process of liquid crystalline CNTFs. Thus, densification

of CNTF in this study is mainly based on phase separation by solubility difference between CSA and acetone.

Advancements

1. The significant improvement of mechanical properties: the world-strongest fiber

The most noticeable achievement of this work is the high-strength of our optimized CNTFs. We started with the as-spun CNTF whose strength is already very high. Note that the final strength reported in the Carbon paper by Cho et al. is 2.19 N/tex, which is similar to the strength of our as-spun CNTF. In general, an effect of a post-treatment is likely to be amplified when the properties of as-spun CNTFs are relatively low, because in these CNTFs, the room for the improvement is large. When the synthesis is well optimized, so the as-spun CNTFs are of high quality as in this work, efficiency of a treatment may not be so obvious. Nevertheless, the CSA treatment nearly doubled specific tensile strength and nearly quadrupled specific tensile modulus (from 48.3 ± 7.4 N/tex to 187.5 ± 7.4 N/tex) of CNTFs that had high I_G/I_D . The results indicate that the CSA densification can effectively maximize the mechanical properties of CNTFs as long as as-spun CNTF has a low defect density.

2. Suggesting the possibility of continuous process

We optimized the immersion time of CNTF in CSA bath and revealed that optimal immersion time is dependent on the intrinsic properties of CNTs. In the case of as-spun CNTFs, the optimal immersion time is in the scale of ~ 1 min, followed by quick coagulation process. The combination of short optimal immersion and coagulation time provides opportunity for continuous production of densified CNTF.

Finally, we revisited the reviewer's previous comments and answered the questions as below. We really appreciate the comments from the reviewer and sincerely hope that this revision would satisfy the reviewer #1.

Answers to the previous comments raised by the reviewer #1

Q1. It will be helpful if the authors can further clarify the correlation between alignment and the improved mechanical properties. For example, the oxidized CNTF by heating and plasma treatment has lower alignment as compared to CNTF without oxidation, but display greater specific tensile strength. This is confusing.

A1. First, we confirmed a clear positive correlation between alignment and mechanical properties. To study the effect of alignment on the mechanical properties, we performed a series of experiments where the stretching ratio was varied (Fig. 3a-c). Given the same immersion time in CSA (1 min), the increased stretching ratio led to increase of alignment and specific tensile strength. But, from this experiment, we also confirmed that the improvement in the density of CNTFs during CSA densification process led to the improvement in the specific strength of CNTFs. Therefore, the improvement of the mechanical properties of CNTF after CSA densification can be explained by the interplay between the alignment and density improvement.

As the reviewer pointed out, the heat-CSA-CNTFs have lower alignment as compared to CSA-CNTF. The final specific tensile strength of heat-CSA-CNTF could be either lower or higher depending on the I_G/I_D . When the I_G/I_D was low, heat-CSA-CNTFs had higher specific tensile strength than CSA-CNTFs. On the contrary, when the I_G/I_D was high, heat-CSA-CNTFs had lower specific tensile strength than CSA-CNTFs.

To explain this, we need to consider the change of density as well as the change of alignment. In the case of low I_G/I_D , heat-CSA-CNTF has a significantly higher density than CSA-CNTF

(Fig. 6d), which compensates for the lower alignment of heat-CSA-CNTF and results in overall improvement of specific tensile strength. On the contrary, in the case of high I_G/I_D , the density of CSA-CNTF is already much higher than that of as-spun CNTF and heat-CSA-CNTF had a minor improvement compared to CSA-CNTF (Fig. 6d), leading to a lower specific tensile strength of heat-CSA-CNTF than CSA-CNTF.

Then the question arises as to why the specific tensile strength of heat-CSA-CNTF is dependent on the I_G/I_D of as-spun CNTF. Our hypothesis is that controlling the DOP is important to achieve highest efficiency of the CSA treatment. In the case of low I_G/I_D , oxidation helped increase the DOP, resulting in improved specific tensile strength. However, in the case of high I_G/I_D , the heat-treated CNTFs would have too high DOP for 1 min immersion in CSA, which is not desirable to achieve high specific tensile strength as discussed in Fig. 3d-f. To confirm the hypothesis and compare the result with that in Fig. 3d-f, we performed CSA treatment with shorter immersion time (0.5 min) using heat-treated CNTF that had medium I_G/I_D (Fig. 6d). As a result, stronger CNTF was obtained with shorter immersion time, indicating that the optimal immersion time for heat-treated medium- I_G/I_D CNTF was shorter than 1 min, which was the optimal immersion time for medium- I_G/I_D CNTF in Fig. 3d-f. The result, combined with the result in Fig. 3d-f, suggests that the optimal immersion time leading to optimal DOP is dependent on the properties of CNTs comprising the CNTF. The answer to this question was clearly reflected in the newly added sections with newly added Fig. 3 and Fig. 6 in the revised manuscript.

Q2. Additionally, different crystallinities of CNTF were tested, and it shows that the oxidation will actually improve the better property for low IG/ID. Will oxidation alter the crystallinity and IG/ID? How do different crystallinities impact the degree of oxidation and

later DOP during CSA. A correlation between crystallinity after oxidation treatment with DOP and mechanical properties will be helpful to establish this trend.

A2. The mechanism by which oxidation improved the efficiency of CSA treatment for low I_G/I_D CNTF is that oxidation improved the solubility of CNTs in CSA and thus promoted protonation and increased DOP in the same time of CSA immersion. It was not because oxidation altered the I_G/I_D of CNTs. This is described in detail in the answer to the above question. We also confirmed that the oxidation by heat treatment in air did not significantly alter the crystallinity by Raman spectroscopy. The result was included in Fig. S8 in the Supplementary Information. In addition, we tried to quantify the O/C ratio of CNTFs with different I_G/I_D after the heat treatment in air by X-ray photoelectron spectroscopy (XPS). Although XPS is not a thorough method to quantify the O/C ratio, it gives a rough evaluation of O/C ratio. The O/C atomic ratios of CNTFs with low and medium I_G/I_D after the heat treatment were 0.179 and 0.225, respectively. From this result, it would be difficult to consider that the increased strength of heat-CSA-CNTF in the case of low I_G/I_D is directly due to higher degree of oxidation. The X-ray photoelectron spectroscopy survey spectra were included in Fig. S9 in the Supplementary Information.

Inserted sentence (the 5th sentence of the 1st paragraph on page 18):

We first confirmed that the heat treatment in air did not significantly alter the defect density (Fig. S8) and that the ratio of oxygen to carbon (O/C) after the heat treatment in air did not vary significantly among CNTFs with different I_G/I_D s (Fig. S9).

Inserted figure (Fig. S8):

Fig. S8. The change of I_G/I_D of CNTFs with two levels of initial I_G/I_D s after heat treatment in air.

Inserted figure (Fig. S9):

Fig. S9. X-ray photoelectron spectroscopy (XPS) survey spectra of CNTFs that had (a) low and (b) medium initial I_G/I_D after heat treatment in air.

(2) Answers to the comments raised by the reviewer #2

Q1. p. 2. l. 22 remove ‘entirely’ and substitute ‘potentially’. The authors have not demonstrated a continuous process embodying the CSA step.

A1. We are sorry for causing such confusion. We substituted ‘entirely’ with ‘potentially’ in the revised manuscript.

Previous sentence (the 3rd sentence of the 1st paragraph on page 2):

Here, we demonstrate the feasibility of highly efficient and entirely continuous fiber-spinning method from the synthesis of carbon nanotubes (CNTs) to the fabrication of highly densified and aligned carbon nanotube fibers (CNTFs) within one-minute processing time.

Revised sentences (from the 3rd sentence to the 4th sentence of the 1st paragraph on page 2):

Here, we demonstrate the feasibility of a highly-efficient and potentially-continuous fiber-spinning method to produce high-performance carbon nanotube (CNT) fiber (CNTF). The processing time is < 1 min from synthesis of CNTs to fabrication of highly densified and aligned CNTFs.

Q2. p. 5. l. 91 insert new sentences possibly using this place to convey other information about the process which does not appear here or in ~SOM. One sentence should mention the nature of the ceramic used in the reactor tube and its diameter, the other should indicate how

the reactor volume was sealed to permit fibre egress. Somewhere else it was hinted that the fibre was drawn out through a water bath. The significance is that the water, if indeed that is what was used, also condenses the fibre, and must be the reason why the directly spun fibre showed the degree of condensation seen in Fig 4. There are other places this information could appear, but it should be in the paper proper, not the SOM.

A2. According to the reviewer's comment, we inserted new sentences to convey the information about tube reactor and the use of water bath for sealing. The ceramic tube used for the synthesis of CNTF was the alumina tube, the inner diameter and length of which were 85 mm and 1,800 mm, respectively. Actually, the diameter and length were mentioned already in the previously revised manuscript (page 6, line 100), but in the revised manuscript, this was deleted and mentioned in a new sentence as below. The reactor was sealed by a water bath, and the synthesized CNTF was drawn through the water bath. Of course, CNTF was initially condensed while passing through the water bath. This is also mentioned in the revised manuscript.

Previous sentence (the 1st sentence of the 1st paragraph on page 5):

To synthesize high-strength as-spun CNTFs, we optimized the direct spinning conditions efficiently by fixing the total H₂ flow rate at 1,200 sccm.

Revised sentences (from the 1st sentence to the 4th sentence of the 1st paragraph on page 5):

To synthesize high-strength as-spun CNTFs, we performed floating catalyst CVD (FC-CVD) using a vertical alumina tube reactor that had inner diameter of 85 mm and length of 1,800 mm (Supplementary Information). The reactor was sealed by a water bath. Synthesized hollow CNT socks transformed to condensed fibers when they were drawn through the water bath. We optimized the direct-spinning conditions by

fixing the total H₂ flow rate at 1,200 sccm.

Previous sentence (the 5th sentence of the 1st paragraph on page 5):

We roughly estimated the ratio of buoyancy force to viscous force which determines whether the backflow occurs in our vertical direct spinning system (inner diameter 85 mm; length 1,800 mm)²² and confirmed that most of our direct spinning conditions were within the range in which the backflow occurs.

Revised sentences (the 1st sentence of the 1st paragraph on page 6):

We roughly estimated the ratio of buoyancy force to viscous force which determines whether the backflow occurs in our vertical direct spinning system²² and confirmed that most of our direct spinning conditions were within the range in which the backflow occurs.

Q3. p. 6. 1. 106. How does ferrocene work in a bubbler, as it sublimes rather readily?

A3. We apologize the reviewer for the confusion. Yes, the ferrocene readily sublimes at the temperature of 65 °C (ferrocene vapor pressure: 21.7 Pa for 60 °C, 93.2 Pa for 80 °C (Appl. Phys. A (2009) 94: 853-860), which is always set for the ferrocene container when CNTF is synthesized, and H₂ gas carries ferrocene molecules into the reactor. To avoid the confusion, we replaced the term ‘bubbler’ with the term ‘container’ in case of the ferrocene in the revised manuscript.

Previous sentence (the 2nd sentence of the 2nd paragraph on page 6):

The ferrocene and thiophene were supplied into the system through separate bubblers, in flowing H₂.

Revised sentence (the 2nd sentence of the 2nd paragraph on page 6):

The temperatures were 65 °C in the ferrocene container and -20 °C in the thiophene bubbler, and both catalyst precursors were supplied into the system by flowing H₂.

Q4. p. 6. Ls. 109-112 I cannot understand this sentence, or even guess at what the authors are trying to convey. Could they please rewrite it, or split it into two or three sentences perhaps?

A4. We appreciate the reviewer for this comment. We started the optimization process with the fixed ferrocene flow rate (0.18 mg/min) and we observed that by decreasing the relative ratios of thiophene and CH₄ with respect to the ferrocene flow rate, we could synthesize CNTFs consisting of decreased numbers of walls, which is in accordance with earlier reports. However, with the fixed ferrocene flow rate (0.18 mg/min), we could spin CNTFs mainly consisting of DWCNTs. To synthesize CNTFs mainly consisting of SWCNTs, we tried to further reduce the relative ratios of thiophene and CH₄ with respect to the ferrocene flow rate (0.18 mg/min). However, this was not successful, because further reduction in the relative ratios of thiophene and CH₄ failed to yield the amount of CNTs that is required to form CNT socks. Therefore, to synthesize CNTFs that consisted mainly of single-walled CNTs (SWCNTs), we increased the ferrocene flow rate to 0.3 mg/min when we reduced the relative ratios of thiophene and CH₄. According to the reviewer's request, we rewrite the previous sentences to clearly convey what we meant in the revised manuscript.

Previous sentence (the 5th sentence of the 2nd paragraph on page 6):

However, we could not spin CNTFs that consisted mostly of single-walled CNTs (SWCNTs) by decreasing the thiophene and CH₄ flow rates with the ferrocene flow rate of 0.18 mg/min; to synthesize CNTFs consisting of SWCNTs, we increased the ferrocene flow rate to 0.3 mg/min.

Revised sentences (from the 5th sentence to the 6th sentence of the 2nd paragraph on

page 6):

However, with the fixed ferrocene flow rate (0.18 mg/min), we could only spin CNTFs that consisted mostly of double-walled CNTs (DWCNTs), even though the thiophene and CH₄ flow rates were decreased; further reduction in the relative ratios of thiophene and CH₄ failed to yield the amount of CNTs that is required to form CNT socks. Therefore, to synthesize CNTFs that consisted mainly of single-walled CNTs (SWCNTs), we increased the ferrocene flow rate to 0.3 mg/min when we reduced the relative ratios of thiophene and CH₄.

Q5. p. 9. l. 153 The paper lacks a proper discussion of the relation of the Raman I_g/I_d ratio to ‘defects’, and it is suggested that it should be inserted here. I think it is generally held that I_d comes from regions close to edge termination of the graphitic layers. There are defects possible in CNTs, such as 5-7 pairs which may not leave a significant imprint on I_d, yet in kinking the tube may severely compromise its ability to be well aligned. On the other hand graphitic particles, possibly associated with ‘blind’ iron particles, but not belonging to the nanotubes which are being aligned may contribute strongly to I_d. I do not think that these comments invalidate the ratio measurements, but there does need to be a discussion of the meaning of this ratio, with the added comment that it will not be affected by lack of perfect alignment of the CNTs.

A5. We thank the reviewer for the helpful suggestion. In the revised manuscript, we added the discussion about various factors in as-spun CNTFs contributing the intensity of the disorder peak in the Raman spectra of CNTFs, which may or may not affect the solvation of CNTs composing CNTF in CSA. Nevertheless, since our as-spun CNTF synthesized in the optimized conditions consists of mainly CNTs (73.8 wt%) and relatively small amount of

carbonaceous impurities (9.4 wt%) (Fig. 2e, Table 1), we speculate that the correlation between the defect density, measured by I_G/I_D from the Raman spectra of CNTFs and the solubility of CNTs composing CNTFs in CSA can be still valid. In addition, we further clarified the reason why we synthesized CNTFs with controlled I_G/I_D .

Previous sentences (from the 1st sentence to the 2nd sentence of the 2nd paragraph on page 9):

The defect density of CNTs critically affects the solubility of CNTs in CSA as side wall protonation, the main mechanism of solvation of CNTs by CSA, is dependent on the ability to share electrons⁵. To investigate how the defect density of CNTs affects the densification process, we synthesized three CNTFs that had nearly identical tensile properties, but different defect density typically represented by I_G/I_D .

Revised sentences (from the 1st sentence of the 2nd paragraph on page 9 to the 1st sentence of the 1st paragraph on page 10):

By principle, the efficiency of the CSA densification process should be highly dependent on the degree to which CSA solvated the CNTs in the CNTFs. CNTs are solvated by side-wall protonation by CSA^{5,25,26} so the speed at which CSA penetrates the CNTFs can be strongly affected by the defect density of CNTs in them. To investigate how the defect density of CNTs in CNTF affects the densification process, we synthesized three CNTFs that had nearly identical tensile properties, but different defect density typically represented by I_G/I_D from the Raman spectra of CNTFs. The intensity I_D of the disorder peak in Raman spectra of CNTFs can be contributed by defects (e.g., kinks, Stone-Wales defects, and sp^3 -hybridization) in CNTs that break the symmetry and perfection of sp^2 -hybridized carbon network, and by carbonaceous impurities such as graphitic shells or particles in CNTF, which would not critically affect the solvation of CNTs in CNTF. However, the as-spun CNTFs synthesized in the

optimized conditions consist mainly of CNTs (73.8 wt%) with a relatively small amount of carbonaceous impurities (9.4 wt%) (Fig. 2e, Table 1), so the correlation between the defect density (I_G/I_D from the Raman spectra of CNTFs) and the solubility of CNTs composing CNTFs in CSA remains valid.

Q6. p. 11. 1. 192 it would add to the impact of the paper, if, after the value for the specific modulus for T1000G, one for specific strength was also added.

A6. According to the reviewer's comment, we added the specific tensile strength of T1000G and compared the specific tensile strength and modulus of our best CSA-CNTF with those of T1000G.

Previous sentence (the 6th sentence of the 1st paragraph on page 11):

The maximum specific tensile strength reached 4.44 N/tex.

Revised sentence (the 6th sentence of the 1st paragraph on page 13):

The maximum specific tensile strength and modulus reached 4.44 N/tex and 195 N/tex.

Previous sentence (the 8th sentence of the 1st paragraph on page 11):

In addition, the specific tensile modulus of CSA-CNTF is superior to those of most commercialized high-strength carbon fibers such as T1000G, which has the specific tensile modulus of 163 N/tex²⁶.

Revised sentence (the 8th sentence of the 1st paragraph on page 13):

In addition to the high specific strength, the specific tensile modulus of CSA-CNTF is superior to those of most commercialized high-strength carbon fibers such as T1000G, which has specific tensile modulus = 163 N/tex and the specific tensile strength = 3.54 N/tex²⁷.

Q7. p. 11. l. 193 The paper requires another significant addition, and this might be the best place. We are invited to note qualitatively that from Figs 4a and c that the cross sectional area is reduced on CSA treatment, even though as mentioned earlier the tex only changed negligibly. Firstly the Figs 4 a-d do not reproduce well enough to be convincing in support of the text. I would strongly suggest that they are printed considerably larger. However, more than so, the paper here is crying out for some numbers as to the changes in volumetric density on CSA treatment. In the SOM the authors describe in some detail how they measured the fibre cross sectional area, so at the very least they should divide the actual values of the tex measurements by the measured cross section areas to give values for the volumetric density which can then be averaged over all the samples. However, this is not all. After CSA treatment the fibre is put into acetone, now acetone is a most effective condensation agent in its own right, so the question raises its head as to how much of the reduction in area is due to the CSA stage and how much due to the acetone. One might guess that it is largely due to the CSA stretch stage, but good science demands that it is checked. The authors should measure cross sectional areas both before and after an acetone treatment but without the CSA stage.

A7. We checked the cross-sectional areas and density of the CNTF before and after acetone treatment without the CSA stage. No densification effect was observed in our additional experiment.

Inserted sentences (from the 1st sentence to the 2nd sentence of the first paragraph on page 15):

To exclude the possibility that the densification can be partly induced by the acetone treatment, we also analyzed the cross-section and the density of CNTF before and after acetone treatment without the CSA stage; we observed no densification effect

(Fig. S5). Thus, the increase of density can be purely attributed to the CSA treatment process.

Inserted Figure (Fig. S5):

Fig. S5. Effect of acetone treatment. Cross-sectional SEM images of CNTFs (a)-(b) before and (c)-(d) after immersion in acetone for 1 min and drying in air. The scale bars in (a) and (c) are 5 μm and those in (b) and (d) are 1 μm .

Q8. p. 13 1.227 Fig 4 (see above)

A8. According to the previous comment, we considerably enlarged Fig. 4a-d and added the calculated values of volumetric densities in Fig. 4b and d. We provided the detailed explanation about how to measure the cross-sectional areas of CNTFs and calculated the volumetric densities of CNTFs in the Supplementary Information.

Previous figure (Fig. 5):

Revised figure (Fig. 6):

Q9. p. 14 l. 251. ‘Unprecedentedly’ and ‘unique’ are tautologous, and either word is a little OTT. I would suggest both words are substituted by just ‘special’.

A9. We thank the reviewer for this indication. Those words were simply substituted by ‘special’ in the revised manuscript.

Previous sentence (the 4th sentence of the 1st paragraph on page 14):

However, our optimum CNTF has extremely high specific tensile strength (4.44 N/tex) as well as a sufficiently high specific electrical conductivity (2,270 S·m²/kg), which makes our CNTF unprecedentedly unique.

Revised sentence (the 5th sentence of the 1st paragraph on page 21):

However, our optimum CNTF has extremely high specific tensile strength (4.44 N/tex) as well as a sufficiently high specific electrical conductivity (2,270 S·m²/kg), which makes our CNTF special.

Q10. p. 15. l. 259 Fig 5. Even at a fibre production rate of 15 m/min (eventual scale up may lead to speeds 10x faster !) an immersion of 1 min would require a bath 15m long. Fig 5 is misleading in this respect. I suggest that both baths are split in two along their vertical bisector, with jagged edges to remind the reader that the actual baths needed will be somewhat longer.

A10. We appreciate the reviewer for the detailed suggestion. In the proposed continuous process, the bath size should be determined by the optimal immersion time as well as the spinning rate. The optimal immersion time as well as the spinning rate should vary depending on the scale-up process and also the schematic simply intended to convey rough concept, not detailed idea. Therefore, we worried that the schematic with too detailed information may also mislead the readers. Instead, we added the specific comment about the scale of bath in the revised manuscript to prevent the readers' misunderstanding, instead of enlarging the number and size of the bath in the scheme. We believe that the added comment would effectively prevent the readers' misunderstanding.

Inserted sentence (the 3rd sentence of the 2nd paragraph on page 21):

The optimal immersion time is dependent on the property of as-synthesized CNTFs, so the length and design of the CSA bath should be determined based on the property of synthesized CNTFs as well as the optimized spinning rate of the scaled-up equipment.

We are so grateful to the reviewers for taking the time to provide us with this review. We feel that the revised version of this manuscript is much better suited for publication in the *Nature Communications* following this review.

REVIEWERS' COMMENTS:

Reviewer #1 (Remarks to the Author):

The authors has made substantial changes in the revised version by including new data to improve the fundamental understanding of the CSA process. The concerns of the reviewer were appropriately addressed in the response letter and also the revised manuscript. I support the publication of this manuscript in Nature Communications considering significant advancements in the CSA process and exceptional mechanical properties achieved for the CNF fibers.